# An active learning framework for multi-group mean estimation

**Abdellah Aznag**     **Rachel Cummings**     **Adam N. Elmachtoub**
Department of Industrial Engineering and Operational Research
Columbia University
`{aa4683, rac2239, ae2516}` `@columbia.edu`

## Abstract

We consider a problem with multiple groups whose data distributions are unknown, and an analyst would like to learn the mean of each group. We consider an active learning framework to sequentially collect $T$ samples with bandit feedback, each period observing a sample from a chosen group. After observing a sample, the analyst may update their estimate of the mean and variance of that group and choose the next group accordingly. The analyst's objective is to dynamically collect samples to minimize the $p$-norm of the vector of variances of the mean estimators after $T$ rounds. We propose an algorithm, Variance-UCB, that selects groups according to a an upper bound on the variance estimate adjusted to the $p$-norm chosen. We show that the regret of Variance-UCB is $\tilde{O}(T^{-2})$ for finite $p$, and prove that no algorithm can do better. When $p$ is infinite, we recover the $\tilde{O}(T^{-1.5})$ regret bound obtained in [4, 13] with improved dependence on the remaining parameters and provide a new lower bound showing that no algorithm can do better.

## 1   Introduction

Obtaining accurate estimates from limited labeled data is a fundamental challenge. To tackle this issue, active learning has emerged as a promising solution framework where a decision-maker strategically selects one sample at a time [35, 15, 22, 40, 24]. The estimation task becomes more complex when faced with a large population of different groups, where it is important that all groups are represented in the estimation procedure. If this allocation is not done correctly, the decision might incur structural bias against some groups [32, 21, 33, 29]. If allowed to adjust the sampling allocation strategy, the analyst can address their biases and re-allocate their sampling resources. A key challenge is to dynamically collect data from different groups while maintaining a reasonable representation of all the groups, which is the problem we tackle in this paper. We provide an active learning framework for dynamic data collection with bandit feedback for estimating the means of multiple groups in a representative manner.

We consider a population partitioned into multiple groups, each with data points drawn from an unknown data distribution, and an analyst would like to learn the mean of each group. Each group's distribution has an unknown mean and unknown variance. At each time period, the analyst collects one sample from a group of their choice and observes a sample from that group. Since only one group is observed, this is also known as bandit feedback. Upon observation, the analyst exploits their new knowledge to optimize their choice of the next group to observe. At the end of the time horizon $T$, the analyst forms a mean estimate for each group. The objective of the analyst is to return a vector of mean estimates with smallest $p$-norm of the variance vector of the mean estimates. The choice of $p$-norm is motivated by its ability to capture different aspects of multi-group estimation performance [10, 23, 28].

37th Conference on Neural Information Processing Systems (NeurIPS 2023).

By considering various values of $p$, we gain insights into different dimensions of estimation accuracy, ranging from the overall spread of the estimates ($p = 2$, Euclidean norm) to the worst case deviation from the true mean ($p = +\infty$, infinity norm), a case that is studied in [4, 13]. This approach allows us to assess the estimation quality from multiple perspectives, providing a more nuanced understanding of how the diverse population is represented in the final estimation of the group means. The analyst can choose $p$ in a way to capture notions of fairness in the data collection process.

## 1.1 Summary of contributions

In this work, we first present an active learning framework that captures the trade-off between representation and accuracy in the task of multi-group estimation. Our framework encapsulates a large class of norms ($p$-norms), which bring various insights into different dimensions of estimation accuracy. In Section 3, we present Variance-UCB, an algorithm that selects groups according to an upper bound on the variance estimate adjusted to the $p$-norm.

We provide a norm-dependent regret bound on Variance-UCB under the two assumptions sub-Gaussian feedback and positive variances. Specifically, we show that the regret is $\tilde{O}(T^{-2})$ for finite $p$ norms (Theorem 1). For the case of the infinite norm, we show that the regret is $\tilde{O}(T^{-1.5})$ (Theorem 3). While bounds at this rate are already established in [4, 13], our bound provides tighter dependencies in the problem parameters $G$ and $\sigma_{\min}$. In the case of the infinite norm, we improve the coefficient of the $T^{-1.5}$ term in the regret from $\sigma_{\min}^{-2} G^{2.5}$ to $G^{1.5}$. This improvement not only tightens the dependency in the number of groups, but it also partially solves one of the open questions left in [13] regarding the role of $\sigma_{\min}$ in the regret bound. While we prove that $\sigma_{\min}$ can still impact the regret, it appears only in lower order terms with respect to $T$.

The analysis and proofs of these upper bounds is presented in Section 4. Our proof technique differs from proofs of similar upper bounds in the related literature. Instead of studying the regret function directly, we first consider its Taylor expansion, and focus our analysis on its dominant term. This technique has two advantages. First, the dominant term is much simpler to analyze (linear in the case of the infinite norm, and quadratic in the case of finite $p$ norms) than the full regret function. Second, the resulting upper bound is tighter in its dominant term, in that it does not suffer from large numerical constants, as is common in the existing literature on regret analysis of bandit algorithms (e.g., [4, 13]).

We also provide new matching lower bounds in $T$ for both finite (Theorem 2) and infinite $p$-norms (Theorem 4), establishing that Variance-UCB achieves the best possible regret in both regimes. These bounds are summarized in Table 1. Prior to our work and to the best of our knowledge, no lower bounds were known for this problem. The analysis of these results is presented in Section 5.

Table 1: Summary of main results

| Norm | Variance-UCB Regret | Lower Bound on Regret for Any Policy |
|---|---|---|
| $p < \infty$ | $\tilde{O}\left(T^{-2}\right)$ (Theorem 1) | $\Omega\left(T^{-2}\right)$ (Theorem 2) |
| $p = \infty$ | $\tilde{O}(T^{-1.5})$ (Theorem 3) | $\Omega(T^{-1.5})$ (Theorem 4) |

We empirically validate our findings by numerical experiments presented in Section 6, which show that our theoretical regret bounds match empirical convergence rates in both cases of finite and infinite $p$-norms. We also provide examples showing that for finite $p$-norms, the smallest variance affects the regret, even when the feedback is Gaussian. This is in contrast to the case of the infinite norm, where it is proven [13] that under Gaussian feedback, the algorithm is not affected by the smallest variance.

## 1.2 Related work

Our motivation stems from growing attention to data collection methods [34, 2, 18, 20]. We focus on the problem of mean estimation and dynamically collecting data to achieve this goal. While there is a substantial body of literature on data acquisition [9, 26], specifically in the presence of privacy concerns and associated costs [19, 30, 31, 12, 17, 16], our approach differs as we do not consider the costs of sharing data. Instead, we concentrate on the data collection process itself, rather than the costs to data providers.

Our work is related to multi-armed bandits problems [5, 37], in the sense that each group can be seen as an arm, and choosing a group to sample from at each time step corresponds to choosing which arm to pull. However, the performance criterion for multi-armed bandits is measured by the difference between the mean of the chosen arm and the best arm [3, 14, 38]. In our framework, the means of the chosen arms do not impact the performance. It is their variances that matter in the optimal solution, as we measure the performance by considering the $p$-norm of the variance of the estimator, which can be non-convex. Because of this, and to the best of our knowledge, usual bandits algorithms and proof techniques [25] do not apply. Instead, we propose Variance-UCB, an algorithm that uses high probability upper bounds on the variance estimates adjusted to the chosen norm.

Considering data acquisition from the perspective of active learning is a natural approach [6, 7, 27], and in that sense our work is also related to active learning [15, 22, 24, 40]. In [1], the authors address the optimal data acquisition problem under the assumption of additive objective functions. They formulate the problem as an online learning problem and leverage well-understood tools from online convex optimization [8, 11, 36]. However, their ideas do not apply to our setting due to the non-additive nature of our decisions over time.

The case where the chosen norm is $\|\cdot\|_\infty$ is introduced in [4], where the authors devise the *GAFS-MAX* algorithm and show that for bounded feedback with known upper bounds, it achieves regret $\tilde{O}(T^{-1.5})$. This case is also studied in [13], where the authors extend the feedback to sub-Gaussian, and devise the *B-AS* algorithm, which also has $\tilde{O}(T^{-1.5})$ regret. One property that emerges in their study is a regret bound deteriorates with $\sigma_{\min}^{-1}$, where $\sigma_{\min}^2$ is the smallest variance across groups. This is counter-intuitive, as smaller variances in general make the learning simpler. The authors pose as an open question whether any algorithm can have performance independent of $\sigma_{\min}$ and show that in the special case of exactly Gaussian feedback, one can derive a $\sigma_{\min}$ free bound. We partially answer this question, by showing that the leading term of the regret bound (with respect to $T$) does not depend on $\sigma_{\min}$. In other words, while $\sigma_{\min}$ can still impact the regret, its effect appears only in lower order terms. Our regret analysis of Variance-UCB for $\|\cdot\|_\infty$ recovers the same dependence in $T$ as in [13], but also has improved dependence in the number of groups $G$ and the variance vector $\boldsymbol{\sigma}$. While [4, 13] serve as a starting point, most of their findings are only applicable to the special case $p = +\infty$. In particular, they cannot be expanded to other choices of norm. We show that for finite $p$ norms, we prove the regret bound is $\tilde{O}(T^{-2})$, which is fundamentally different than the previous $\tilde{O}(T^{-1.5})$ bound.

## 2 Model

We consider a population partitioned into $G$ disjoint groups. Each group is represented by an index $g$ from $[G] := \{1, \ldots, G\}$. Each individual in the population holds a real-valued data point; data from each group $g \in [G]$ are distributed according to an unknown distribution $\mathcal{D}_g$, with unknown mean $\mu_g$ and unknown variance $\sigma_g^2$. We denote $\sigma_{\min} := \min_{g \in [G]} \sigma_g$ and let $\boldsymbol{\mathcal{D}} := \mathcal{D}_1 \times \ldots \times \mathcal{D}_G$. The analyst wishes to compute an unbiased estimate of the population mean for each group over $T$ times of data collection, sampling only one group at a time. The set of feasible policies is defined as

$$\Pi := \left\{ \boldsymbol{\pi} = \{\pi_t\}_{t \in [T]} \mid \pi_t \in G^{t-1} \times \mathbb{R}^{t-1} \to \Delta(G), \ \forall t \in [T] \right\},$$

where $\Delta(G)$ is the set of measures supported on $[G]$. Let $n_{g,T}$ denote the number of collected samples from group $g$ at the end of time $T$, and let $\hat{\mu}_{g,T}$ be the sample mean estimator of $\mu_g$ for $n_{g,T}$ collected samples. Once all data have been collected at the end of the time horizon $T$, the analyst will compute the sample mean of each group, i.e.,

$$\hat{\mu}_{g,T} = \frac{1}{\sum_{t=1}^{T} \mathbb{1}_{X_t=g}} \sum_{t=1}^{T} \mathbb{1}_{X_t=g} Y_t.$$

Note that the vector $\hat{\boldsymbol{\mu}}_T$ is always an unbiased estimator of the vector $\boldsymbol{\mu}$, as long as the policy $\boldsymbol{\pi}$ samples at least once from each group.

The variance of $\hat{\mu}_{g,T}$ is $\frac{\sigma_g^2}{n_{g,T}}$. The $p-$norm of the vector of variances of $n_{g,T}$ is denoted as

$$R_p(\boldsymbol{n}_T) := \left\| \left\{ \frac{\sigma_g^2}{n_{g,T}} \right\}_{g=1}^{G} \right\|_p. \tag{1}$$

The analyst wishes to minimize $\mathbb{E}[R_p(\boldsymbol{n}_T)]$.

When choosing a policy, the analyst does not have access to the true standard deviation vector $\boldsymbol{\sigma} := (\sigma_1, \ldots, \sigma_G)$, which is needed to compute the value $R_p(\boldsymbol{n}_T)$. Therefore the analyst must learn $\boldsymbol{\sigma}$ through their decisions. We benchmark the performance of a policy against the best possible performance in a complete information setting where $\boldsymbol{\sigma}$ is known, that is,

$$\min_{\boldsymbol{n} \in \mathbb{N}^G} R_p(\boldsymbol{n}) \quad s.t. \quad \sum_{g \in [G]} n_g = T. \tag{2}$$

The optimization program (2) can be difficult to solve and analyze due to the integer constraints. Instead of using the solution to (2) as a benchmark, we use the solution to its continuous relaxation (a lower bound on (2)), which we denote as,

$$R_p^* := \min_{\boldsymbol{n} \in \mathbb{R}_+^G} R_p(\boldsymbol{n}) \quad s.t. \quad \sum_{g \in [G]} n_g = T. \tag{3}$$

Thus, we can define the regret of a policy as

$$\text{Regret}_{p,T}(\boldsymbol{\pi}, \boldsymbol{\mathcal{D}}) := \mathbb{E}_\pi[R_p(\boldsymbol{n}_T)] - R_p^*. \tag{4}$$

For our analysis, we assume that the distributions $\mathcal{D}_g$ are sub-Gaussian, as stated in Assumption 1, with corresponding constants $c_1, c_2$ known to the analyst.[1]

**Assumption 1** (Sub-Gaussianity). *For each $g \in [G]$, $\mathcal{D}_g$ is sub-Gaussian. That is, there exist universal constants $c_1, c_2 > 0$ such that for any $\epsilon > 0$,*

$$\mathbb{P}_{Y \sim \mathcal{D}_g}\left(|Y - \mu_g| \geq \epsilon\right) \leq c_1 \exp\left(-\epsilon^2 / c_2\right).$$

*We assume that such $c_1, c_2$ are known to the analyst.*

Moreover, we assume that all groups have some variation in their data, as stated in Assumption 2.

**Assumption 2** (Positive Variance). *The minimum group variance is positive; i.e., $\sigma_{\min} = \min_{g \in [G]} \sigma_g > 0$.*

## 3 Our algorithm: Variance-UCB

Our algorithm, Variance-UCB, builds an increasingly accurate upper confidence bound for each $\sigma_g$, which we denote $\text{UCB}_t(\sigma_g)$. Recall that $n_{g,t}$ denotes the number of collected samples from group $g$ through time $t$. At each time $t$, $\sigma_g$ can be estimated via the sample standard deviation

$$\hat{\sigma}_{g,t} := \sqrt{\frac{1}{n_{g,t} - 1} \sum_{s \leq t : X_s = g} (Y_s - \hat{\mu}_{g,t})^2}.$$

For convention, we set $\hat{\sigma}_{g,0} = \hat{\sigma}_{g,1} = +\infty$, indicating that at least two samples are required to obtain a meaningful estimate of $\sigma_g$. We can then define

$$\text{UCB}_t(\sigma_g) := \hat{\sigma}_{g,t} + \frac{C_T}{\sqrt{n_{g,t}}}, \tag{5}$$

where

$$C_T := 2\sqrt{2 \log\left(2T^4\right) c_1 \log(c_2 T^4)} + \frac{2\sqrt{c_1 \log\left(2T^4\right)\left(1 + c_2 + \log(c_2 T^4)\right)}}{(1 - T^{-4})\sqrt{2 \log(2T^4)}} T^{-2}. \tag{6}$$

$C_T$ was introduced in [13], and captures the trade-off between accuracy of the upper confidence bound and confidence in the estimate, and is a polylogarithmic factor in $T$. In particular, $C_T$ can be constructed by the analyst since it depends only on $c_1$, $c_2$, and $T$, which are known.

At time $t + 1$, the algorithm chooses the next group $X_{t+1}$ using the rule:

$$X_{t+1} = \arg\max_{g \in [G]} \frac{\text{UCB}_t(\sigma_g)^{\frac{2p}{p+1}}}{n_{g,t}},$$

---

**Algorithm 1** Variance-UCB $(p, T, G, c_1, c_2)$

---

**Input:** norm parameter $p$, time horizon $T$, number of groups $G$ and subgaussian parameters $c_1, c_2$.

1: Initialize $n_{g,0} = 0, \hat{\sigma}_{g,0} = \hat{\sigma}_{g,1} = +\infty, \forall g \in [G]$
2: Compute $C_T$ according to (6).
3: **for** $t = 0, \ldots, T-1$ **do**
4:     Compute $\text{UCB}_t(\sigma_g)$: $\text{UCB}_t(\sigma_g) = \hat{\sigma}_{g,t} + \frac{C_T}{\sqrt{n_{g,t}}}, \quad \forall g \in [G]$
5:     Select group $X_{t+1} = \text{argmax}_g \frac{\text{UCB}_t(\sigma_g)^{\frac{2p}{p+1}}}{n_{g,t}}$
6:     Observe feedback $Y_{t+1} \sim \mathcal{D}_{X_{t+1}}$
7:     Update the number of samples: $n_{g,t+1} = n_{g,t} + \mathbb{I}_{X_{t+1}=g}, \quad \forall g \in [G]$
8:     Update the mean estimates: $\hat{\mu}_{g,t+1} = \frac{1}{n_{g,t+1}} \sum_{s=1}^{t+1} \mathbb{1}_{X_s=g} Y_s, \quad \forall g \in [G]$
9:     Update the standard deviation estimates: $\hat{\sigma}_{g,t+1} :=$ $\sqrt{\frac{1}{n_{g,t+1}-1} \sum_{s \leq t+1: X_s=g} (Y_s - \hat{\mu}_{g,t+1})^2}, \quad \forall g \in [G]$
10: **end for**

**Output:** $\hat{\mu}_{g,T} = \frac{1}{n_{g,T}} \sum_{s=1}^{T} \mathbb{1}_{X_s=g} Y_s, \quad \forall g \in [G]$

---

where where ties in the argmax can be broken arbitrarily. The analyst then observes a new sample $Y_{t+1}$ from the chosen group $X_{t+1}$ and updates the upper confidence bounds accordingly. Variance-UCB is presented formally in Algorithm 1.

Variance-UCB takes as input the time horizon $T$, the norm $p$, the groups $[G]$, and the sub-Gaussian parameters $c_1, c_2$. The algorithm initializes the upper confidence bound for every group to be infinity, which reflects the absence of knowledge of $\{\sigma_g\}$.

In the special case $p = +\infty$, Variance-UCB (instantiated with a different choice of $C_T$) coincides with the *B-AS* algorithm of [13].[2]

### 3.1 Regret guarantees

Our first main result gives theoretical bounds on the performance of Algorithm 1 for finite $p \in [1, \infty)$. We show in Theorem 1 that when $p$ is finite, Variance-UCB incurs regret of $\tilde{O}(T^{-2})$.

**Theorem 1.** *For any $\mathcal{D}$ that satisfies Assumptions 1 and 2 and for any finite $p$, the regret of Variance-UCB is at most $\tilde{O}(T^{-2})$. That is,*

$$Regret_{p,T}(\text{Variance-UCB}, \mathcal{D}) = \tilde{O}(T^{-2}).$$

Our second main result, Theorem 2, provides a matching lower bound for our problem. Thus showing that the performance of Variance-UCB and its analysis is the best possible in terms of $T$.

**Theorem 2.** *Let $p$ be finite and $\kappa$ be a universal constant. For any online policy $\pi$, there exists an instance $\mathcal{D}_{\pi}$ such that for any $T \geq 1$,*

$$Regret_{p,T}(\pi, \mathcal{D}_{\pi}) \geq \kappa(p+1)T^{-2} + O\left(T^{-2.5}\right) = \Theta(T^{-2}).$$

Our analysis and proof techniques can be extended naturally to the case where $p = +\infty$. However, the $\tilde{O}(T^{-2})$ regret no longer holds, and the convergence rate jumps to $\tilde{O}(T^{-3/2})$.

**Theorem 3.** *Let $\Sigma_{\infty} := \sum_{g \in [G]} \sigma_g^2$. For any $\mathcal{D}$ that satisfies Assumptions 1 and 2,*

$$Regret_{\infty,T}(\text{Variance-UCB}, \mathcal{D}) \leq \left(C_T \sqrt{\Sigma_{\infty}} + C_T^2\right) G^{1.5} T^{-1.5} + o(T^{-1.5}) = \tilde{O}(T^{-1.5}).$$

Note that a similar bound for Theorem 3 has already been established in [13]:

$$\text{Regret}_{\infty,T}(\text{B-AS}, \mathcal{D}) \leq \frac{28230(c_1((c_2+2)^2+1)\log^2(T)\Sigma_{\infty}G^{2.5}}{\sigma_{\min}^2 T^{1.5}} + o\left(T^{-1.5}\right).$$

---

[1]In Section 6, we empirically evaluate the impact of mis-estimating $c_1$ and $c_2$.

[2]Our results still hold under this modified parameter setting for any $C_T$ satisfying Lemma 2.

Our result in Theorem 3 improves the existing regret in all the problem parameters, i.e., $\Sigma_\infty$ to $\sqrt{\Sigma_\infty}$, and from $G^{2.5}$ to $G^{1.5}$). The most significant improvement lies in removing the dependence on $\sigma_{\min}$ in the main term of the regret. While $\sigma_{\min}$ still appears in the negligible term $o(T^{-1.5})$, this term will be asymptotically dominated for even moderate $T$. This improved bound gives a better understanding on how $\sigma_{\min}$ impacts the performance of the algorithm, which was left as an open question in [13].

Finally, we give a matching lower bound for the case when $p = \infty$, showing that the analysis of Variance-UCB is tight in $T$ when $p = +\infty$. To the best of our knowledge, no lower bound for this problem was previously known.

**Theorem 4.** *For any online policy $\boldsymbol{\pi}$, there exists an instance $\mathcal{D}_{\boldsymbol{\pi}}$ such that for any $T \geq 1$,*

$$Regret_{\infty,T}(\boldsymbol{\pi}, \mathcal{D}_{\boldsymbol{\pi}}) \geq \frac{1}{2} G^{1.5} T^{-1.5}.$$

In Sections 4 and 5, we give outlines of the proofs of Theorems 1 and 2, respectively. While we briefly mention why both results change at $p = +\infty$, the full proofs for Theorems 3 and 4 is deferred to Appendix C.

## 4 Deriving the upper bounds

In this section, we give an overview of the main steps to prove Theorem 1. A complete proof is given in Appendix A. We first show in Lemma 1 that the optimal $R_p^*(\boldsymbol{\sigma})$ is achieved for an optimal static policy $\boldsymbol{n}_T^*(p) = \{n_{g,T}^*(p)\}_{g \in [G]}$ that assumes knowledge of $\boldsymbol{\sigma}$ and samples each group $g$ proportionally to $\sigma_g^{\frac{2p}{p+1}}$.

**Lemma 1.** *[Benchmark analysis] For each $t \in \mathbb{N}^*$ and $p \in [1, +\infty]$, let $n_{g,t}^* = \dfrac{\sigma_g^{\frac{2p}{p+1}} t}{\sum_{h \in [G]} \sigma_h^{\frac{2p}{p+1}}}$. Then,*

$$R_p^*(\boldsymbol{\sigma}) = R_p(\boldsymbol{n}_T^*) = \frac{1}{T} R_p(\boldsymbol{n}_1^*).$$

The proof of Lemma 1 utilizes the KKT conditions of the optimization program (3) that defines $R_p^*$. For convenience, we introduce[3] $\Sigma_p := \sum_{g \in [G]} \sigma_g^{\frac{2p}{p+1}}$ for each $p \in [1, +\infty]$ so that we can simplify $n_{g,t}^* = \frac{t}{\Sigma_p} \sigma_g^{\frac{2p}{p+1}}$. Using Equation (4), the expression of regret can be simplified to

$$\text{Regret}_{p,T}(\boldsymbol{\pi}, \mathcal{D}) = \mathbb{E}_{\boldsymbol{\pi}} \left[ R_p(\boldsymbol{n}_T) - R_p(\boldsymbol{n}_T^*) \right]. \tag{7}$$

From Eq. (7), we can understand the behavior of the regret by answering the following questions.

1. How close is the random variable $\boldsymbol{n}_T$ to the optimal sampling vector $\boldsymbol{n}_T^*$?
2. How does the curvature of $R_p(.)$ affect the difference $R_p(\boldsymbol{n}_T) - R_p(\boldsymbol{n}_T^*)$?

**Upper bounding the error $\boldsymbol{n}_T - \boldsymbol{n}_T^*$:** Before we answer the first question, we show in Lemma 2 that $\text{UCB}_t(\sigma_g)$ is an increasingly accurate upper bound for $\sigma_g$.

**Lemma 2.** *For all $g \in [G]$, with probability at least $1 - \tilde{O}(T^{-2})$,*

$$0 \leq UCB_t(\sigma_g)^{\frac{2p}{p+1}} - \sigma_g^{\frac{2p}{p+1}} \leq \frac{4C_T}{\sqrt{n_{g,t}}} \frac{p}{p+1} \left( \sigma_g + \frac{2C_T}{\sqrt{n_{g,t}}} \right)^{\frac{p-1}{p+1}}.$$

The proof of Lemma 2 utilizes Assumption 1 and the choice of $\text{UCB}_t(\sigma_g)$[4], as defined in Eq. (5).

Using Lemma 2, the difference betwen the number of samples Variance-UCB collects and the optimal number of samples can be bounded with high probability, which we state in Lemma 3.

---

[3]Note that the definition of $\Sigma_p$ is consistent with the definition of $\Sigma_\infty$ introduced in Theorem 3

[4]Lemma 2 holds for all choices of $p \in [1, +\infty]$. In particular, it is still true in the case where $p = +\infty$, as it can be understood by taking the limit $p \to +\infty$ in the inequalities: $0 \leq \text{UCB}_t(\sigma_g)^2 - \sigma_g^2 \leq \frac{4C_T}{\sqrt{n_{g,t}}} \left( \sigma_g + \frac{2C_T}{\sqrt{n_{g,t}}} \right)$.

**Lemma 3.** *Variance-UCB collects a vector of samples $\boldsymbol{n}$ such that for all $g \in [G]$, with probability at least $1 - \tilde{O}(T^{-2})$,*

$$n_{g,T} - n_{g,T}^* \leq 3 + \frac{4C_T p}{\Sigma_p(p+1)} \left( \sigma_g + \frac{2C_T}{\sqrt{n_{g,T}^*}} \right)^{\frac{p-1}{p+1}} \sqrt{n_{g,T}^*} = \Theta(\sqrt{T}).$$

To understand the meaning of Lemma 3, we note that the feedback-dependent structure of the algorithm makes the numbers of samples $n_{g,T}$ correlated across groups and over time, since the algorithm attempts to sample regularly from all groups. A key technical challenge is to decouple $n_{g,t}$ across groups and derive an instance dependent upper bound on $\boldsymbol{n}_T - \boldsymbol{n}_T^*$. This challenge does not arise in the classic multi-armed bandits setting, where the decision-maker's goal is to repeatedly pull only the best arm.

**Curvature properties of $R_p$ around the optimal value $\boldsymbol{n}_T^*$:** To answer the second question, we exploit the smoothness of $R_p(\cdot)$ around $\boldsymbol{n}_T^*$ to approximate it with a polynomial function of $\boldsymbol{n}$. Since $\boldsymbol{n}_T^*$ is the minimizer of $R_p(\cdot)$ (subject to $\sum_{g \in [G]} n_g = T$) and $R_p(\cdot)$ is differentiable around $\boldsymbol{n}_T^*$, the first order of the Taylor approximation of $R_p$ vanishes as $T$ grows large, formalized in Lemma 4.

**Lemma 4.** *Let $p < +\infty$ and $\boldsymbol{n}' \in \mathbb{R}_+^G$ such that $\sum_{g \in [G]} n_g' = T$. Then,*

$$\left| \frac{R_p(\boldsymbol{n}') - R_p(\boldsymbol{n}_T^*)}{R_p(\boldsymbol{n}^*)} - \frac{p+1}{2} \sum_{g \in [G]} \frac{(n_g' - n_{g,T}^*)^2}{T n_{g,T}^*} \right| \leq \frac{7(p+2)^2 \Sigma_p^2}{\sigma_{\min}^2} \max_g \left( \frac{n_{g,T}^*}{n_g'} \right)^{3p+3} \frac{\|\boldsymbol{n}' - \boldsymbol{n}_T^*\|_\infty^3}{T^3}.$$

We note that the bound in Lemma 4 holds regardless of the choice of the vector $\boldsymbol{n}'$, including those generated by Variance-UCB. One can interpret the upper bound in Lemma 4 as follows: the term $\frac{p+1}{2} \sum_{g \in [G]} \frac{(n_g' - n_{g,T}^*)^2}{T n_{g,T}^*}$ represents the exact Taylor first-order approximation of $\frac{R_p(\boldsymbol{n}') - R_p(\boldsymbol{n}_T^*)}{R_p(\boldsymbol{n}^*)}$, and the right hand side represents an upper bound on this approximation. In particular, assuming an error $\boldsymbol{\epsilon} = \boldsymbol{n}' - \boldsymbol{n}^*$, Lemma 4 can be restated as,

$$\frac{R_p(\boldsymbol{n}') - R_p(\boldsymbol{n}_T^*)}{R_p(\boldsymbol{n}^*)} = \Theta\left( \frac{\|\boldsymbol{\epsilon}\|^2}{T^2} \right) + o\left( \frac{\|\boldsymbol{\epsilon}\|^2}{T^2} \right).$$

**Putting everything together:** The rest of the proof consists of applying Lemmas 3 and 4 to the regret expression in Eq. (7). By Lemma 3, with high probability, $\|\boldsymbol{n}_T - \boldsymbol{n}_T^*\|_\infty = \tilde{O}\left( \sqrt{T} \right)$. Applying this to Lemma 4 gives that $R_p(\boldsymbol{n}_T) - R_p(\boldsymbol{n}_T^*) = \|\boldsymbol{n}_T - \boldsymbol{n}_T^*\|_\infty^2 R_p^* \cdot O\left( T^{-2} \right)$. By Lemma 1, $R_p^* = \Theta(T^{-1})$, and therefore with high probability, $R_p(\boldsymbol{n}_T) - R_p(\boldsymbol{n}_T^*) = \tilde{O}(T^{-2})$. With additional work, we show that the equality also holds in expectation, which implies from Eq. (7) that $\text{Regret}_{p,T}(\text{Variance-UCB}, \mathcal{D}) = \mathbb{E}_\pi[R_p(\boldsymbol{n}) - R_p(\boldsymbol{n}_T^*)] = \tilde{O}(T^{-2})$.

## 4.1 The case $p = +\infty$

Even though Lemmas 1, 2, and 3 hold for $p = +\infty$, the approximation guarantee given in Lemma 4 does not hold because $R_\infty$ is not differentiable at $\boldsymbol{n}_T^*$. As an alternative, we derive a first order upper bound, which we state in Lemma 5.

**Lemma 5.** *Let $\boldsymbol{\sigma} \in \mathbb{R}_+^G$ and $\boldsymbol{n}' \in \mathbb{R}_+^G$ such that $\sum_{g \in [G]} n_g' = T$. Then,*

$$\frac{R_\infty(\boldsymbol{n}_T') - R_\infty(\boldsymbol{n}_T^*)}{R_\infty(\boldsymbol{n}_T^*)} \leq -\min_g \left( \frac{n_g'}{n_{g,T}^*} - 1 \right) + \frac{1}{4} \max_g \left( \frac{n_g'}{n_{g,T}^*} - 1 \right)^2 \max_g \left( \frac{n_{g,T}^*}{n_g'} \right)^3$$

Similar to Lemma 4, we interpret the upper bound above by decomposing it into a main term, $-\min_g \left( \frac{n_g'}{n_{g,T}^*} - 1 \right)$, and an error term, $\frac{1}{4} \max_g \left( \frac{n_g'}{n_{g,T}^*} - 1 \right)^2 \max_g \left( \frac{n_{g,T}^*}{n_g'} \right)^3$. As opposed to the case where $p < +\infty$, $R_\infty$ is not differentiable in $\boldsymbol{n}^*$, thus the inequality above does not stem from a Taylor approximation. Adapting the bound in Lemma 5 in the same steps as in **Step 3** gives a regret bound of $\tilde{O}(T^{-1.5})$.

# 5 Deriving the lower bounds

In this section, we provide the key ideas behind the proof of Theorem 2, with the full proof given in Appendix B. Given any policy $\pi$, the idea is to pick $\mathcal{D}_\pi$ from two constructed instances $\mathcal{D}^a = \mathcal{D}_1^a \times \cdots \times \mathcal{D}_G^a$ and $\mathcal{D}^b = \mathcal{D}_1^b \times \cdots \times \mathcal{D}_G^b$. The interaction of $\mathcal{D}^a$ (resp. $\mathcal{D}^b$) with $\pi$ yields a random vector of the number of collected samples, denoted by $n^a$ (resp. $n^b$). $\mathcal{D}^a$ and $\mathcal{D}^b$ must satisfy the following two conflicting properties:

1. $\mathcal{D}^a$ and $\mathcal{D}^b$ are sufficiently similar, in the sense that the distributions of $n^a$ and $n^b$ are close, so that $\pi$ would have a hard time distinguishing between $\mathcal{D}^a$ and $\mathcal{D}^b$. This notion of similarity is captured by the KL-divergence of $\mathcal{D}^a$ and $\mathcal{D}^b$.

2. $\mathcal{D}^a$ and $\mathcal{D}^b$ are also sufficiently dissimilar in the sense that they induce distinct optimal allocation rules $n_a^*$ and $n_b^*$. Specifically, any allocation $n$ cannot be simultaneously close to both $n_a^*$ and $n_b^*$, and $n$ incurs a high regret under at least one of the two instances. This notion of dissimilarity is captured below.

Let $\sigma^a \in \mathbb{R}_+^G$ be the vector of standard deviations of all groups under $\mathcal{D}^a$, and let $S_\epsilon^a$ be the set of allocations $n$ such that $R_p(n; \sigma^a)$ is within $\epsilon$ of $R_p^*(\sigma^a)$ for any $n \in S_\epsilon^a$. Formally,

$$S_\epsilon^a := \left\{ n \in \mathbb{N}^G \,\middle|\, \sum_g n_g = T, \quad R_p(n, \sigma^a) - R_p^*(\sigma^a) \leq \epsilon \right\}.$$

Let $\sigma^b$ be a second vector and define $S_\epsilon^b$ similarly. We define the dissimilarity $d(\sigma^a, \sigma^b) := \inf_{\epsilon \geq 0} \{ S_\epsilon^a \cap S_\epsilon^b \neq \emptyset \}$, which is the smallest $\epsilon$ so that these sets of allocations are disjoint. There is a tension between the two requirements on $\mathcal{D}^a$ and $\mathcal{D}^b$, which is formalized next in Lemma 6.

**Lemma 6.** *Let $\pi$ be a fixed policy and $\mathcal{D}^a$, $\mathcal{D}^b$ be two instances with standard deviation vectors $\sigma^a, \sigma^b$, respectively. Then,*

$$\max\{Regret_{p,T}(\pi, \mathcal{D}^a), Regret_{p,T}(\pi, \mathcal{D}^b)\} \geq d(\sigma^a, \sigma^b) \exp\left( -\sum_{g \in [G]} \mathbb{E}_{\pi, \mathcal{D}^a}[n_{g,t}] KL(\mathcal{D}_g^a || \mathcal{D}_g^b) \right).$$

The proof follows from an adapted version of LeCam's method [39]. To understand the implication of Lemma 6, we examine the two terms involved in the right hand side of the inequality. The first term, $d(\sigma^a, \sigma^b)$, increases the further $\sigma^a$ and $\sigma^b$ are from each other. The second term has the opposite monotonicity, as it decreases with the $KL$ divergence of the two instances. These conflicting terms capture the trade-off of loss minimization versus information gain, and deriving the lower bound will come from constructing two instances for which this trade-off is maximised.

Specifically, $\mathcal{D}^a$ is chosen such that all groups have data distributed according to $\mathcal{D}_g^a \sim \mathcal{N}\left(0, \sigma^2\right)$; we denote $\sigma^a := \sigma(1, \ldots, 1)$ as the vector of standard deviations of each group. Even though instance $a$ has all identical groups, the policy $\pi$ may treat groups differently, and the interaction of $\pi$ with $\mathcal{D}^a$ may yield give different expected numbers of samples from each group. We pick $h \in [G]$ such that $\mathbb{E}_{\pi, \mathcal{D}^a}[n_{h,T}]$ is minimized over $[G]$, and construct $\mathcal{D}^b$ as follows:

$$\mathcal{D}^b : \begin{cases} \mathcal{D}_h^b \sim \mathcal{N}\left(0, \sigma^2(1 + \sqrt{\frac{G}{T}})\right) \\ \mathcal{D}_g^b \sim \mathcal{N}\left(0, \sigma^2\right), \quad \forall g \neq h \end{cases}.$$

On the one hand, notice that $\mathcal{D}^a$ and $\mathcal{D}^b$ only differ in their coordinate $h$, so that $KL(\mathcal{D}_g^a || \mathcal{D}_g^b) = \mathbb{1}(g = h)\Theta(GT^{-1})$, and by minimality of $h$, $\mathbb{E}_{\pi, \mathcal{D}^a}[n_{h,t}] \leq \frac{T}{G}$, so that

$$\sum_{g \in [G]} \mathbb{E}_{\pi, \mathcal{D}^a}[n_{g,t}] KL(\mathcal{D}_g^a || \mathcal{D}_g^b) = \mathbb{E}_{\pi, \mathcal{D}^a}[n_{h,t}] KL(\mathcal{D}_h^a || \mathcal{D}_h^b) \leq \frac{T}{G}\Theta(GT^{-1}) = \Theta(1).$$

Second, we show that $d(\sigma^a, \sigma^b) = \Theta(T^{-2})$. We do this by measuring the distance between the sets $S_\epsilon^a$ and $S_\epsilon^b$ for a fixed $\epsilon$, and then estimate the smallest $\epsilon$ for which this distance is 0. This smallest $\epsilon$ corresponds to $d(\sigma^a, \sigma^b)$. Combining both in Lemma 6 yields $\max\{\text{Regret}_{p,T}(\pi, \mathcal{D}^a), \text{Regret}_{p,T}(\pi, \mathcal{D}^b)\} \geq \Theta(T^{-2})$.

**Remark 1.** *For $p = +\infty$, the proof remains the same except that in this case $d(\sigma^a, \sigma^b) = \Theta(T^{-3/2})$, resulting in an overall lower bound of $\Theta(T^{-3/2})$. See Appendix C.3 for the complete proof.*

## 6 Numerical study

In this section, we present experimental results on the empirical performance of Variance-UCB. We explore the impact of varying each important parameter of the problem: the time horizon $T$, norm parameter $p$, number of groups $G$, distributions $\mathcal{D}_g$, and the sub-Gaussianity parameters $c_1, c_2$. In all the experiments $\mathcal{D}_g$ follow Gaussian distributions. Except where they are varied, the default parameter settings are $T = 10^5$, $p = 2$, $G = 2$, with the respective data distributions of groups 1 and 2 as $\mathcal{N}(1, 1)$ and $\mathcal{N}(2, 2.5)$, satisfying $c_1 = c_2 = 5$. For each parameter setting evaluated, we run Variance-UCB 500 times and report average regret over all 500 runs. For convenience, time and regret are presented on logarithmic scales.

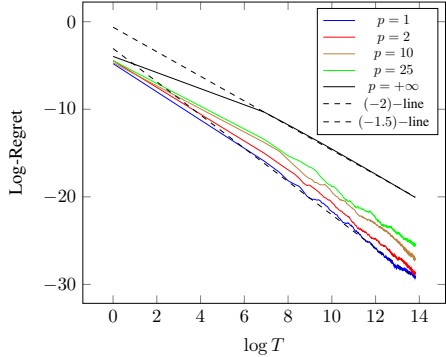

Figure 1: Impact of the $p$-norm on the regret.

First, we vary the choice of the parameter $p \in \{1, 2, 10, 25, +\infty\}$ and observe its effect on the regret. Figure 1 shows that the convergence rates for each $p$ precisely match those predicted by Theorems 1 and 3: a slope of -2 for the finite values of $p$ ($\{1, 2, 10, 25\}$), and a slope of -1.5 for $p = \infty$.

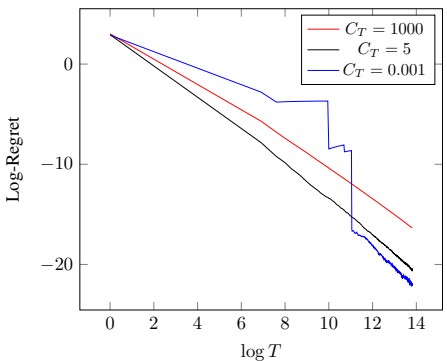

Figure 2: Impact of mis-estimating $C_T$ on the regret

Next, we study how mis-specifying the sub-Gaussianity parameters $c_1$ and $c_2$ affect the regret of Algorithm 1. The parameter values $(c_1, c_2)$ only affect the algorithm's behavior through $C_T$, so it is more natural to directly study the impact of errors in $C_T$. For the parameters used in our experimental setup, the value of $C_T$ prescribed in Equation (6) is $\sim 5$; we consider both underestimating and overestimating $C_T$, and evaluate regret when instead plugging in values of $C_T \in \{0.001, 5, 1000\}$ in the UCB update step defined in Equation (5). Figure 2 shows that choosing an overestimate of $C_T$ incurs an increased regret. This is not surprising since choosing a larger $C_T$ decreases the confidence of the algorithm in its estimates, and forces over-exploration. Choosing a low $C_T$ also incurs a higher regret which can more severely impact the performance, as the algorithm under-explores and can get stuck in sub-optimal behavior. However, with a long enough time horizon, the algorithm will eventually estimate $\sigma$ accurately, regardless of the small choice of $C_T$. As observed in Figure 2, the smallest value of $C_T = 0.001$ initially has the highest regret, corresponding to incorrect estimates and insufficient exploration, and then finally converges very late to have the lowest regret. Even for very large $T$, this curve has higher variance due to the noise in the estimates.

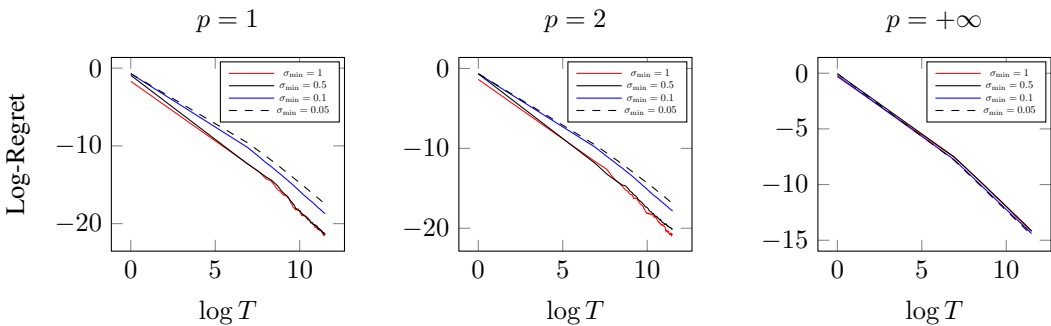

Figure 3: The impact of $\sigma_{\min}$ on the regret

Next, we study the effect of varying the lowest variance $\sigma_{\min}^2 \in \{0.05, 0.1, 0.5, 1\}$ while holding all other variances fixed, for each $p \in \{1, 2, +\infty\}$. First, Figure 3 shows that varying $\sigma_{\min}$ has no effect on the regret when $p = +\infty$. This matches a result of [13], where they prove that when $p = +\infty$, their UCB-style algorithm (very similar to Variance-UCB) is not affected by the lowest variance when the feedback is Gaussian. However, our experimental results show that this phenomenon does not persist when $p$ is finite, as illustrated in Figure 3, where we observe that regret decreases when the lowest variance $\sigma_{\min}$ increases. This is surprising since increasing the lowest variance makes the feedback more volatile, and one would expect an increase in regret as a result.

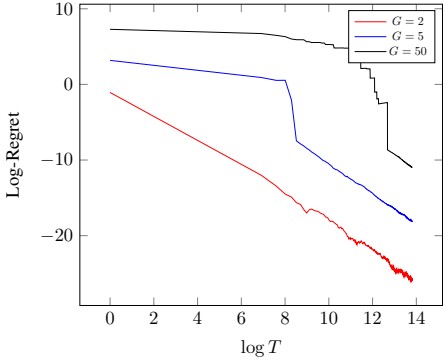

Figure 4: Impact of the number of groups $G$ on regret

Finally, we vary the number of groups $G \in \{2, 10, 50\}$. For the additional groups, we generate their data from Gaussian $\mathcal{D}_g$, with means $\mu_g \sim \mathcal{U}([-1, 1])$ and standard deviations $\sigma_g \sim \mathcal{U}([2, 4])$ independently for each group. From Figure 4, we observe that the regret increases in the number of groups, as expected. When the number of groups is small ($G = 2$), Variance-UCB quickly enters a regime where regret decreases quickly. However, as the number of groups grows ($G \in \{10, 50\}$), the necessary time to enter the decay regime increases ($\sim 30,000$ for $G = 10$ and $\sim 90,000$ for $G = 50$). Initially the algorithm samples (on average) uniformly across all groups due to the UCB term outweighing the sample variance estimates, and each group must wait to be sampled enough times for the algorithm to estimate its optimal sampling rate. This delay will naturally increase with the number of groups. Once the confidence bounds are small enough, the algorithm samples the highest variances first. This causes abrupt variation (especially in the case where $p$ is small) because the objective function is very sensitive to changes in one coordinate.

## Acknowledgements

The authors gratefully acknowledge the support of National Science Foundation grants IIS-2147361. The authors would also like to thank the reviewers for their valuable feedback which helped improve the paper.

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

# A  Proof of Theorem 1

## A.1  Proof of Lemma 1

The proof of Lemma 1 consists of using optimality conditions on $r_p$:

**Lemma 1.** *[Benchmark analysis] For each $t \in \mathbb{N}^*$ and $p \in [1, +\infty]$, let $n_{g,t}^* = \dfrac{\sigma_g^{\frac{2p}{p+1}} t}{\sum_{h \in [G]} \sigma_h^{\frac{2p}{p+1}}}$. Then,*

$$R_p^*(\boldsymbol{\sigma}) = R_p(\boldsymbol{n}_T^*) = \frac{1}{T} R_p(\boldsymbol{n}_1^*).$$

*Proof.* The case of $p = +\infty$ is discussed in [4, 13]. We focus on when $p < +\infty$. Recall the definition of $R_p^*$ from Eq. (3):

$$R_p^* := \min_{\boldsymbol{n} \in \mathbb{R}_+^G} R_p(\boldsymbol{n}) \quad s.t. \quad \sum_{g \in [G]} n_g = T.$$

Since the function $x \to x^p$ is increasing in $\mathbb{R}_+$, we can replace the objective by $[R_p(\boldsymbol{n})]^p = \sum_{g \in [G]} \frac{\sigma_g^{2p}}{n_g^p}$. Any feasible point $\boldsymbol{n}$ with a zero coordinate would have $R_p(\boldsymbol{n}) = +\infty$, therefore any argmin to the optimization program $\boldsymbol{n}^*$ above must have positive coordinates, and satisfies the KKT conditions

$$\forall g \in [G], \quad \frac{\partial}{\partial n_g} R_p(\boldsymbol{n})^p - \frac{\partial}{\partial n_g} \lambda \left( \sum_{h \in [G]} n_h - T \right) = 0,$$

$$\lambda \in \mathbb{R}.$$

For each $g \in [G]$, the first line of the system above is equivalent to $-p \frac{\sigma_g^{2p}}{n_g^{p+1}} - \lambda = 0$. Therefore the KKT conditions imply that

$$\frac{\sigma_1^{2p}}{n_1^{p+1}} = \ldots = \frac{\sigma_G^{2p}}{n_G^{p+1}}, \quad \text{or equivalently,} \quad \frac{\sigma_1^{\frac{2p}{p+1}}}{n_1} = \ldots = \frac{\sigma_G^{\frac{2p}{p+1}}}{n_G} = \frac{\sum_{g \in [G]} \sigma_g^{\frac{2p}{p+1}}}{\sum_{g \in [G]} n_g} = \frac{\sum_{g \in [G]} \sigma_g^{\frac{2p}{p+1}}}{T}.$$

Hence the unique minimizer is the vector

$$\boldsymbol{n}_T^* = \frac{T}{\sum_{g \in [G]} \sigma_g^{\frac{2p}{p+1}}} \boldsymbol{\sigma}^{\frac{2p}{p+1}}.$$

Therefore, $R_p^* = R_p(\boldsymbol{n}_T^*)$. Moreover, $R_p(\boldsymbol{n}_T^*) = R_p(T\boldsymbol{n}_1^*) = \frac{1}{T} R_p(\boldsymbol{n}_1^*)$, where the last equality follows from the homogeneity of the norm. $\square$

## A.2  Proof of Lemma 2

In this section, we establish a key property of $\text{UCB}_t(\sigma_g)$, which relies on Assumption 1.

**Lemma 2.** *For all $g \in [G]$, with probability at least $1 - \tilde{O}(T^{-2})$,*

$$0 \leq UCB_t(\sigma_g)^{\frac{2p}{p+1}} - \sigma_g^{\frac{2p}{p+1}} \leq \frac{4C_T}{\sqrt{n_{g,t}}} \frac{p}{p+1} \left( \sigma_g + \frac{2C_T}{\sqrt{n_{g,t}}} \right)^{\frac{p-1}{p+1}}.$$

*Proof.* Following the notations of Section 3, we introduce the following event:

$$\mathcal{A}_T := \bigcap_{g \in [G], 2 \leq t \leq T} \left\{ |\hat{\sigma}_{g,t} - \sigma_g| \leq \frac{C_T}{\sqrt{t}} \right\}.$$

Based on corollary 1 of [13], we have

(i) $\mathbb{P}_\pi(\mathcal{A}_T) \geq 1 - 2GT^{-2.5}$,

(ii) conditionally on $\mathcal{A}_T$,

$$\forall g \in [G], t \geq 2G, \quad |\hat{\sigma}_{g,t} - \sigma_g| \leq \frac{C_T}{\sqrt{n_{g,t}}},$$

so that conditionally on $\mathcal{A}_T$, we have:

$$
\begin{aligned}
\mathrm{UCB}_t\left(\sigma_g\right) &= \hat{\sigma}_{g,t} + \frac{C_T}{\sqrt{n_{g,t}}} \\
&= \sigma_g + \frac{C_T}{\sqrt{n_{g,t}}} + \left(\hat{\sigma}_{g,t} - \sigma_g\right) \\
&\in \sigma_g + \frac{C_T}{\sqrt{n_{g,t}}} + \left[-\frac{C_T}{\sqrt{n_{g,t}}}, \frac{C_T}{\sqrt{n_{g,t}}}\right] \\
&= \sigma_g + \left[0, \frac{2C_T}{\sqrt{n_{g,t}}}\right],
\end{aligned}
$$

where the first Eq. is due to the definition of $\mathrm{UCB}_t$ introduced in (5), and the bounding is due to the definition of $\mathcal{A}_T$. Next, notice that the function $x \to (\sigma_g + x)^{\frac{2p}{p+1}}$ is increasing, which implies that

$$\sigma_g^{\frac{2p}{p+1}} \leq \mathrm{UCB}_t\left(\sigma_g\right)^{\frac{2p}{p+1}} \leq \left(\sigma_g + \frac{2C_T}{\sqrt{n_{g,t}}}\right)^{\frac{2p}{p+1}}, \tag{8}$$

which proves the leftmost inequality in Lemma 2. To prove the rightmost inequality, notice that $x \to (\sigma_g + x)^{\frac{2p}{p+1}}$ is also convex, therefore by Jensen's inequality,

$$\left(\sigma_g + \frac{2C_T}{\sqrt{n_{g,t}}}\right)^{\frac{2p}{p+1}} - \sigma_g^{\frac{2p}{p+1}} \leq \frac{2C_T}{\sqrt{n_{g,t}}} \frac{2p}{p+1}\left(\sigma_g + \frac{2C_T}{\sqrt{n_{g,t}}}\right)^{\frac{2p}{p+1}-1} = \frac{4C_T}{\sqrt{n_{g,t}}} \frac{p}{p+1}\left(\sigma_g + \frac{2C_T}{\sqrt{n_{g,t}}}\right)^{\frac{p-1}{p+1}},$$

which concludes the proof. $\qquad\square$

### A.3 Proof of Lemma 3

The goal of this section is to prove Lemma 3, which consists of bounding with high probability $\boldsymbol{n} - \boldsymbol{n}^*$. To do so, we will design an alternative sequence $\tilde{\boldsymbol{n}}$, that is simultaneously easy to analyze, and upper bounds $\boldsymbol{n}$ with high probability. The motivation for the choice of $\tilde{\boldsymbol{n}}$ comes from the relaxation of the choice made at every time step by Variance-UCB.

We assume through the whole section what the event $\mathcal{A}_T$ is realized. For convenience, we view the right hand side of Lemma 2 as a quantity of its own, and introduce the *width* function

$$w_g : x > 0 \to w_g(x) := \frac{4C_T}{\Sigma_p\sqrt{x}} \frac{p}{p+1}\left(\sigma_g + \frac{2C_T}{\sqrt{x}}\right)^{\frac{p-1}{p+1}},$$

so that the inequality in Lemma 2 can be rewritten as

$$0 \leq \mathrm{UCB}_t\left(\sigma_g\right)^{\frac{2p}{p+1}} - \sigma_g^{\frac{2p}{p+1}} \leq \Sigma_p w_g(n_{g,t}). \tag{9}$$

We start by proving the following lemma, which follows from the decisions Variance-UCB makes:

**Lemma 7.** *For $t \geq 2G$, we have:*

$$n_{X_{t+1},t} - tw_{X_{t+1}}(n_{X_{t+1},t}) \leq n^*_{X_{t+1},t}$$

*Proof.* The proof exploits the greedy property of the algorithm. A group needs to be sampled exactly twice to have a finite UCB. Therefore, for $t = 1, \ldots, 2G$, Variance-UCB samples every group twice (in an arbitrary order), so that at $n_{1,2G} = \ldots = n_{G,2G} = 2$ and $\mathrm{UCB}_{2G}(\sigma_1)^{\frac{2p}{p+1}}, \ldots, \mathrm{UCB}_{2G}(\sigma_G)^{\frac{2p}{p+1}} < +\infty$. By choice of $X_{t+1}$, the following inequality holds:

$$\forall g \in [G], \quad \frac{\mathrm{UCB}_t(\sigma_g)^{\frac{2p}{p+1}}}{n_{g,t}} \leq \frac{\mathrm{UCB}_t(\sigma_{X_{t+1}})^{\frac{2p}{p+1}}}{n_{X_{t+1},t}}. \tag{10}$$

On the one hand, from the leftmost inequality in (9),

$$\forall g \in [G], \quad \frac{\sigma_g^{\frac{2p}{p+1}}}{n_{g,t}} \leq \frac{\mathrm{UCB}_t(\sigma_g)^{\frac{2p}{p+1}}}{n_{g,t}}. \tag{11}$$

On the other hand, from the rightmost inequality in (9),

$$\frac{\mathrm{UCB}_t(\sigma_{X_{t+1}})^{\frac{2p}{p+1}}}{n_{X_{t+1},t}} \leq \frac{\sigma_{X_{t+1}}^{\frac{2p}{p+1}} + \Sigma_p w_{X_{t+1}}(n_{X_{t+1},t})}{n_{X_{t+1},t}}. \tag{12}$$

Therefore by combining both Inequalities (11) and (12) in Inequality (10),

$$\forall g \in [G], \quad \frac{\sigma_g^{\frac{2p}{p+1}}}{n_{g,t}} \leq \frac{\sigma_{X_{t+1}}^{\frac{2p}{p+1}} + \Sigma_p w_{X_{t+1}}(n_{X_{t+1},t})}{n_{X_{t+1},t}},$$

which implies, after multiplying both sides by $n_{g,t} n_{X_{t+1},t}$ and summing over $g \in [G]$,

$$n_{X_{t+1},t} \underbrace{\sum_{g \in [G]} \sigma_g^{\frac{2p}{p+1}}}_{=\Sigma_p} \leq \left( \sigma_{X_{t+1}}^{\frac{2p}{p+1}} + \Sigma_p w_{X_{t+1}}(n_{X_{t+1},t}) \right) \underbrace{\sum_{g \in [G]} n_{g,t}}_{=t}.$$

Dividing both sides by $\Sigma_p > 0$, and using the formula $n_{g,t}^* = \frac{\sigma_g^{\frac{2p}{p+1}}}{\Sigma_p} t$ (see Lemma 1) implies

$$n_{X_{t+1},t} \leq n_{X_{t+1},t}^* + t w_{X_{t+1}}(n_{X_{t+1},t}).$$

Lemma 7 follows by substracting $t w_{X_{t+1}}(n_{X_{t+1},t})$ from both sides. $\qquad \square$

Lemma 7 states that the possible excess between the number of samples output by the algorithm and the optimal number of samples is not too big, and can be controlled by the width $w$. Since the width decreases in the number of samples, the function

$$x \rightarrow x - t w_g(x)$$

must be increasing and has therefore an inverse function that is also increasing, which we denote $W_g^t(x)$. We introduce the following sequence, which mimics the behavior stated in Lemma 7:

$$\tilde{n}_{g,t} = n_{g,t} \qquad\qquad\qquad \text{For } t = 1, \ldots, 2G$$
$$\tilde{n}_{g,t+1} = \tilde{n}_{g,t} + \mathbb{1}\left( \tilde{n}_{g,t} \leq W_g^t\left(n_{g,t}^*\right) \right) \qquad\qquad\qquad \text{For } t \geq 2G$$

The sequence is easier to analyze and upper bounds the true number of samples $n$:

**Lemma 8.** $\tilde{n} \geq n$.

*Proof.* By construction, the result holds for $t = 1, \ldots, 2G$. Assume for the sake of contradiction that the result does not hold for a $g \in [G]$ and $t + 1 > 2G$, and take such a $t$ minimal. For such a pair $(g, t)$:

$$\begin{aligned}
1 \geq \mathbb{1}(X_{t+1} = g) = n_{g,t+1} - n_{g,t} \\
> \tilde{n}_{g,t+1} - \tilde{n}_{g,t} \\
= \mathbb{1}(\tilde{n}_{g,t} \leq W_g^t(n_{g,t}^*)) \\
= \mathbb{1}(n_{g,t} \leq W_g^t(n_{g,t}^*)) \geq 0,
\end{aligned}$$

where the first step follows from the definition of $n$. By minimality of $t$, $n_{g,t+1} > \tilde{n}_{g,t+1}$ and $n_{g,t} = \tilde{n}_{g,t}$, which induces the second step. The third step follows from the definition of $\tilde{n}$, and the last step follows once again from the minimality of $t$.

The strict inequality in the chain of inequalities implies that $\mathbb{1}(X_{t+1} = g) = 1$ and $\mathbb{1}(n_{g,t} \leq W_g^t(n_{g,t}^*)) = 0$, so that

$$n_{X_{t+1},t} > W_g^t(n_{X_{t+1},t}^*).$$

By taking the inverse of the increasing function $W_g^t$ on both sides, the previous inequality can be rewritten as

$$n_{X_{t+1},t} - t w_{X_{t+1}}(n_{X_{t+1},t}) > n_{X_{t+1},t}^*,$$

contradicting Lemma 7. Therefore the assumption is wrong and $\tilde{n} \geq n$, which completes the proof. $\qquad \square$

**Lemma 9.** *For a fixed g,the sequence $\{W_g^t(n_{g,t}^*)\}_{t \geq 1}$ is increasing. Consequently,*

$$\tilde{n}_{g,t} \leq W_g^t(n_{g,t}^*)^+ + 2$$

*Proof.* For a fixed $x > 0$ and $t \geq 1$,

$$(x - (t+1)w_g(x)) - (x - tw_g(x)) = -w_g(x) < 0,$$

therefore the sequence of functions $\{x \to x - tw_g(x)\}_{t \geq 1}$ is decreasing in $t$. Consequentely, the sequence of its inverse functions $\{x \to W_g^t(x)\}_{t \geq 1}$ is increasing in $t$:

$$W_g^{t+1}(n_{g,t+1}^*) \geq W_g^t(n_{g,t+1}^*). \tag{13}$$

Moreover, the function $W_g^t$ is increasing in $\mathbb{R}_+$:

$$W_g^t(n_{g,t+1}^*) \geq W_g^t(n_{g,t}^*). \tag{14}$$

By combining Equations (13) and (14), we obtain

$$W_g^{t+1}(n_{g,t+1}^*) \geq W_g^t(n_{g,t}^*),$$

which proves that the sequence $\{W_g^t(n_{g,t}^*)\}_{t \geq 1}$ is increasing, thus completing the proof for the first part of Lemma 9.

The second part follows by induction on $t \geq 1$. For $t \leq 2G$, the result holds immediately as $\tilde{n}_{g,t} = n_{g,t}$ and $n_{g,t} \leq 2$. We assume the result holds for a $t \geq 2G$. We distinguish two cases:

- $\tilde{n}_{g,t} \leq W_g^t\left(n_{g,t}^*\right)$: This implies that:

$$\tilde{n}_{g,t+1} = \tilde{n}_{g,t} + 1$$
$$\leq W_g^t\left(n_{g,t}^*\right) + 1$$
$$\leq W_p^{t+1,g}\left(n_{g,t+1}^*\right) + 1$$
$$\leq W_p^{t+1,g}\left(n_{g,t+1}^*\right)^+ + 2,$$

  where the first step stems from the definition of $\tilde{n}$, the second step stems from the assumption $\tilde{n}_{g,t} \leq W_g^t\left(n_{g,t}^*\right)$, and the third step stems from the first part of the proof.

- $\tilde{n}_{g,t} > W_g^t\left(n_{g,t}^*\right)$: This implies that:

$$\tilde{n}_{g,t+1} = \tilde{n}_{g,t}$$
$$\leq W_g^t\left(n_{g,t}^*\right)^+ + 2$$
$$\leq W_p^{t+1,g}\left(n_{g,t+1}^*\right)^+ + 2,$$

  where the first step stems from the definition of $\tilde{n}$, the second step stems from the induction hypothesis, and the third step stems from the first part of the proof.

Studying both cases concludes the induction and proves the inequality for all $t \geq 1$. This concludes the proof of Lemma 9. □

Next, we derive an upper bound on $W_g^t$:

**Lemma 10.** *For $x > 0$, $W_g^t(x) \leq \frac{x}{1 - t\frac{w_g(x)}{x}}$.*

*Proof.* Let $x, y > 0$ with $0 \leq y - tw_g(y) = x$, so that $y = W_g^t(x)$ by definition of $W_g^t$. we have:

$$\frac{x}{1 - t\frac{w_g(x)}{x}} = \frac{y - tw_g(y)}{1 - t\frac{w_g(y - tw_g(y))}{x - tw_g(x)}}$$

$$= y\frac{1 - t\frac{w_g(y)}{y}}{1 - t\frac{w_g(y - tw_g(y))}{y - tw_g(y)}}$$

$$\geq y = W_g^t(x),$$

where the last step follows from the function $y \to \frac{w_g(y)}{y}$ being decreasing, which concludes the proof. □

We are now ready to prove Lemma 3.

**Lemma 3.** *Variance-UCB collects a vector of samples $\boldsymbol{n}$ such that for all $g \in [G]$, with probability at least $1 - \tilde{O}(T^{-2})$,*

$$n_{g,T} - n^*_{g,T} \leq 3 + \frac{4C_T p}{\Sigma_p(p+1)} \left( \sigma_g + \frac{2C_T}{\sqrt{n^*_{g,T}}} \right)^{\frac{p-1}{p+1}} \sqrt{n^*_{g,T}} = \Theta(\sqrt{T}).$$

*Proof.* Conditionally on $\mathcal{A}_T$, all the previous lemmas stated in this section hold. Consequently,

$$
\begin{aligned}
n_{g,T} &\leq \tilde{n}_{g,T} \\
&\leq 2 + W^t_g(n^*_{g,T})^+ \\
&\leq 2 + \frac{n^*_{g,T}}{1 - T^{\frac{w_g(n^*_{g,T})}{n^*_{g,T}}}} \\
&\leq 2 + n^*_{g,T} + 1 + T w_g(n^*_{g,T}) \\
&= 3 + n^*_{g,T} + \frac{4C_T p}{\Sigma_p(p+1)} \left( \sigma_g + \frac{2C_T}{\sqrt{n^*_{g,T}}} \right)^{\frac{p-1}{p+1}} \sqrt{n^*_{g,T}},
\end{aligned}
$$

where the first step stems from Lemma 8, the second step stems from Lemma 9, the third step stems from first order approximations and the last step stems from Lemma 10. From Assumption 2, $\boldsymbol{\sigma} > 0$, therefore $\boldsymbol{n}^* = \Theta(T)$ and the right hand side is $\Theta(\sqrt{T})$. $\qquad\square$

### A.4 Proof of Lemma 4

The goal of this section is to prove Lemma 4. We will do so by combining the optimality of $\boldsymbol{n}^*$ and the curvature properties of $R_p$.

**Lemma 4.** *Let $p < +\infty$ and $\boldsymbol{n}' \in \mathbb{R}^G_+$ such that $\sum_{g \in [G]} n'_g = T$. Then,*

$$
\left| \frac{R_p(\boldsymbol{n}') - R_p(\boldsymbol{n}^*_T)}{R_p(\boldsymbol{n}^*)} - \frac{p+1}{2} \sum_{g \in [G]} \frac{(n'_g - n^*_{g,T})^2}{T n^*_{g,T}} \right| \leq \frac{7(p+2)^2 \Sigma_p^2}{\sigma^2_{\min}} \max_g \left( \frac{n^*_{g,T}}{n'_g} \right)^{3p+3} \frac{\|\boldsymbol{n}' - \boldsymbol{n}^*_T\|^3_\infty}{T^3}.
$$

*Proof.* Since $R_p$ is defined over the constrained set $\{\boldsymbol{n} \in \mathbb{R}^G_+ | \sum_g n_g = T\}$ with empty interior, Taylor inequality can't be directly applied. To overcome this, we construct an alternative function that has the same curvature as $R_p$ while being defined on a non-empty interior domain. Define

$$\mathcal{K} := \left\{ \boldsymbol{\lambda} \in [0,1]^{G-1} \,\Big|\, \sum_g \lambda_g \leq 1 \right\}. \tag{15}$$

For each vector $\boldsymbol{n} \in \mathbb{R}^G_+$ with $\sum^T_{t=1} n_g = T$, we set for each $g \in [G-1]$ $\lambda_g := \frac{n_g}{T}$. We have $\boldsymbol{\lambda} \in \mathcal{K}$. Moreover, we have from (1):

$$R_p(\boldsymbol{n}) = \left\| \left\{ \frac{\sigma^2_g}{n_g} \right\}^G_{g=1} \right\|_p = \frac{1}{T} \left\| \left\{ \frac{\sigma^2_1}{\lambda_1}, \dots, \frac{\sigma^2_{G-1}}{\lambda_{G-1}}, \frac{\sigma^2_G}{1 - \lambda_1 - \dots - \lambda_{G-1}} \right\} \right\|_p. \tag{16}$$

where the last equality follows from the homogeneity property of the $p$-norm. Eq. (16) motivates the introduction of the re-scaled function $r$:

$$r_p : \boldsymbol{\lambda} \in \mathcal{K} \to r_p(\boldsymbol{\lambda}) := \left\| \left\{ \frac{\sigma^2_1}{\lambda_1}, \dots, \frac{\sigma^2_{G-1}}{\lambda_{G-1}}, \frac{\sigma^2_G}{1 - \lambda_1 - \dots - \lambda_{G-1}} \right\} \right\|_p, \tag{17}$$

so that Eq. (16) can be written as

$$R_p(\boldsymbol{n}) = \frac{1}{T} r_p(\boldsymbol{\lambda}). \tag{18}$$

From the definition of $\mathcal{K}$ in (15), the interior of $\mathcal{K}$, denoted $\mathcal{K}^\circ$, is non-empty and is equal to

$$\mathcal{K}^\circ = \left\{ \boldsymbol{\lambda} \in \mathcal{K} | \forall g \in [G-1], \lambda_g > 0, \quad \sum_{g \in [G-1]} \lambda_g < 1 \right\}.$$

Moreover, the function $r_p$ is $\mathcal{C}^3$ in $\mathcal{K}^\circ$. From Lemma 1, $\boldsymbol{\lambda}^* \in \mathcal{K}^\circ$. Therefore from Taylor's theorem, we have for $\boldsymbol{\lambda} \in \mathcal{K}^\circ$:

$$\left| r_p(\boldsymbol{\lambda}) - r_p(\boldsymbol{\lambda}^*) - (\boldsymbol{\lambda} - \boldsymbol{\lambda}^*)\nabla r_p(\boldsymbol{\lambda}^*) - \frac{1}{2}\langle \mathcal{H}(\boldsymbol{\lambda}^*)(\boldsymbol{\lambda} - \boldsymbol{\lambda}^*), \boldsymbol{\lambda} - \boldsymbol{\lambda}^* \rangle \right| \le \|\boldsymbol{\lambda} - \boldsymbol{\lambda}^*\|_\infty^3 \sup_{\substack{\boldsymbol{u} \in [\boldsymbol{\lambda}, \boldsymbol{\lambda}'] \\ x,y,z \in \mathbb{N} \\ x+y+z=3 \\ \{g,h,i\} \subset [G-1]}} \left| \frac{1}{x!y!z!} \frac{\partial^3 r_p(\boldsymbol{u})}{\partial \lambda_g \partial \lambda_h \partial \lambda_i} \right|,$$

$$\tag{19}$$

where $\nabla$ is the gradient operator and $\mathcal{H}$ is the hessian operator. We exploit the optimality of $\boldsymbol{\lambda}^*$ to derive simple expressions for $\nabla r_p(\boldsymbol{\lambda}^*)$, $\mathcal{H}(\boldsymbol{\lambda}^*)$, and $\frac{\partial^3 r_p(\boldsymbol{u})}{\partial \lambda_g \partial \lambda_h \partial \lambda_i}$. First, since $\boldsymbol{\lambda}^*$ is optimal and an interior point of $\mathcal{K}$, the following equality holds:

$$\nabla r_p(\boldsymbol{\lambda}^*) = 0_G. \tag{20}$$

In particular, $\langle \boldsymbol{\lambda} - \boldsymbol{\lambda}^*, \nabla r_p(\boldsymbol{\lambda}^*) \rangle = 0$. $\langle \mathcal{H}(\boldsymbol{\lambda}^*)(\boldsymbol{\lambda} - \boldsymbol{\lambda}^*), \boldsymbol{\lambda} - \boldsymbol{\lambda}^* \rangle$ is simpified in the following lemma, which proof is deferred to A.6:

**Lemma 11.** $\langle \mathcal{H}(\boldsymbol{\lambda}^*)(\boldsymbol{\lambda} - \boldsymbol{\lambda}^*), \boldsymbol{\lambda} - \boldsymbol{\lambda}^* \rangle = (p+1)r_p(\boldsymbol{\lambda}^*) \sum_{g \in [G]} \frac{(\lambda_g - \lambda_g^*)^2}{\lambda_g^*}$.

so that Inequality (19) is rewritten as

$$\left| r_p(\boldsymbol{\lambda}) - r_p(\boldsymbol{\lambda}^*) - \frac{1}{2}(p+1)r_p(\boldsymbol{\lambda}^*) \sum_{g \in [G]} \frac{(\lambda_g - \lambda_g^*)^2}{\lambda_g^*} \right| \le \|\boldsymbol{\lambda} - \boldsymbol{\lambda}^*\|_\infty^3 \sup_{\substack{\boldsymbol{u} \in [\boldsymbol{\lambda}, \boldsymbol{\lambda}'] \\ x,y,z \in \mathbb{N} \\ x+y+z=3 \\ \{g,h,i\} \subset [G-1]}} \left| \frac{1}{x!y!z!} \frac{\partial^3 r_p(\boldsymbol{u})}{\partial \lambda_g \partial \lambda_h \partial \lambda_i} \right|.$$

$$\tag{21}$$

It remains to bound $\frac{1}{x!y!z!} \frac{\partial^3 r_p(\boldsymbol{u})}{\partial \lambda_g \partial \lambda_h \partial \lambda_i}$. To do this, the following lemma is applied (the proof is also deferred to A.6):

**Lemma 12.** *The following inequality holds:*

$$\sup_{\substack{\boldsymbol{u} \in [\boldsymbol{\lambda}, \boldsymbol{\lambda}'] \\ x,y,z \in \mathbb{N} \\ x+y+z=3 \\ \{g,h,i\} \subset [G-1]}} \left| \frac{1}{x!y!z!} \frac{\partial^3 r_p(\boldsymbol{u})}{\partial \lambda_g \partial \lambda_h \partial \lambda_i} \right| \le \frac{7(p+2)^2}{(\min_g \lambda_g^*)^2} \max_g \left( \frac{\lambda_g^*}{\lambda_g} \right)^{3p+3} r_p(\boldsymbol{\lambda}^*).$$

which implies after dividing both sides by $r_p(\boldsymbol{\lambda})$

$$\left| \frac{r_p(\boldsymbol{\lambda}) - r_p(\boldsymbol{\lambda}^*)}{r_p(\boldsymbol{\lambda}^*)} - \frac{p+1}{2} \sum_{g \in [G]} \frac{(\lambda_g - \lambda_g^*)^2}{\lambda_g^*} \right| \le \frac{7(p+2)^2}{(\min_g \lambda_g^*)^2} \max_g \left( \frac{\lambda_g^*}{\lambda_g} \right)^{3p+3} \|\boldsymbol{\lambda} - \boldsymbol{\lambda}^*\|_\infty^3. \quad (22)$$

By using the change of variable from $\boldsymbol{\lambda}$ to $\boldsymbol{n}'$, we have

$$R_p(\boldsymbol{n}') = \frac{1}{T}r_p(\boldsymbol{\lambda}), \quad R_p(\boldsymbol{n}^*) = \frac{1}{T}r_p(\boldsymbol{\lambda}^*), \quad \|\boldsymbol{\lambda} - \boldsymbol{\lambda}^*\|_\infty \le \frac{\|\boldsymbol{n}' - \boldsymbol{n}^*\|_\infty}{T}$$

which implies

$$\frac{R_p(\boldsymbol{n}') - R_p(\boldsymbol{n}_T^*)}{R_p(\boldsymbol{n}^*)} \le \frac{p+1}{2} \sum_{g \in [G]} \frac{(n_g' - n_{g,T}^*)^2}{Tn_{g,T}^*} + \frac{7(p+2)^2 \Sigma_p^2}{\sigma_{\min}^2} \max_g \left( \frac{n_{g,T}^*}{n_g'} \right)^{3p+3} \frac{\|\boldsymbol{n}' - \boldsymbol{n}_T^*\|_\infty^3}{T^3},$$

hence proving Lemma 5. $\qquad\square$

## A.5 Putting everything together

We are now ready to complete the proof of Theorem 1.

**Theorem 1.** *For any $\mathcal{D}$ that satisfies Assumptions 1 and 2 and for any finite $p$, the regret of Variance-UCB is at most $\tilde{O}(T^{-2})$. That is,*

$$Regret_{p,T}(\textit{Variance-UCB}, \mathcal{D}) = \tilde{O}(T^{-2}).$$

*Proof.* First, notice that

$$
\begin{aligned}
\text{Regret}_{p,T}(\text{Variance-UCB}) &= \mathbb{E}_\pi[R_p(\boldsymbol{n}) - R_p^*] \\
&= \mathbb{E}[R_p(\boldsymbol{n}) - R_p^* | \mathcal{A}_T]\mathbb{P}_\pi(\mathcal{A}_T) + \mathbb{E}[R_p(\boldsymbol{n}) - R_p^* | \mathcal{A}_T^c]\mathbb{P}_\pi(\mathcal{A}_T^c) \\
&\leq \mathbb{E}[R_p(\boldsymbol{n}) - R_p^* | \mathcal{A}_T] + \|\boldsymbol{\sigma}^2\|_p \mathbb{P}_\pi(\mathcal{A}_T^c),
\end{aligned}
$$

where the first step stems from the definition of regret introduced in Eq. (4), the second step stems from the law of total expectation, and the third step stems from both $\mathbb{P}(\mathcal{A}_T) \leq 1$ and $R_p(\boldsymbol{n}) - R_p^* \leq \|\boldsymbol{\sigma}^2\|_p$. It remains to show that each term in the rightmost side is in $\tilde{O}(T^{-2})$. First, $\mathbb{P}_\pi(\mathcal{A}_T^c) \leq 2GT^{-2.5} = \tilde{O}(T^{-2})$. Next, we have conditionally on $\mathcal{A}_T$:

$$
\begin{aligned}
\frac{R_p(\boldsymbol{n}) - R_p(\boldsymbol{n}^*)}{R_p(\boldsymbol{n}^*)} &\leq \frac{p+1}{2} \sum_{g \in [G]} \frac{(n_g - n_{g,T}^*)^2}{Tn_{g,T}^*} + \frac{7(p+2)^2\Sigma_p^2}{\sigma_{\min}^2} \max_g \left(\frac{n_{g,T}^*}{n_g}\right)^{3p+3} \frac{\|\boldsymbol{n} - \boldsymbol{n}^*\|_\infty^3}{T^3} \\
&\leq \frac{p+1}{2} \frac{G\|\boldsymbol{n} - \boldsymbol{n}^*\|_\infty^2}{T\min_g n_{g,T}^*} + \frac{7(p+2)^2\Sigma_p^2}{\sigma_{\min}^2} \max_g \left(\frac{n_{g,T}^*}{n_g}\right)^{3p+3} \frac{\|n - n^*\|_\infty^3}{T^3},
\end{aligned}
$$

where the first inequality stems from Lemma 4, and the second inequality stems from $(n_{g,T} - n_{g,T}^*)^2 \leq \|\boldsymbol{n} - \boldsymbol{n}^*\|_\infty^2$. Since $\sum_g n_{g,T} = T$, from Lemma 3,

$$
\begin{aligned}
n_{g,T} - n_{g,T}^* &= -\sum_{h \neq g} n_{h,T} - n_{h,T}^* \\
&\geq -3(G-1) - \sum_{h \neq g} \frac{4C_T p}{\Sigma_p(p+1)} \left(\sigma_g + \frac{2C_T}{\sqrt{n_{h,T}^*}}\right)^{\frac{p-1}{p+1}} \sqrt{n_{h,T}^*} \\
&\geq -3(G-1) - G \max_h \frac{4C_T p}{\Sigma_p(p+1)} \left(\sigma_g + \frac{2C_T}{\sqrt{n_{h,T}^*}}\right)^{\frac{p-1}{p+1}} \sqrt{n_{h,T}^*} \\
&\geq -3G - \frac{4GC_T p}{\Sigma_p(p+1)} \left(\min \sigma_g + \frac{2C_T}{\sqrt{\min_h n_{h,T}^*}}\right)^{\frac{p-1}{p+1}} \sqrt{\min_h n_{h,T}^*},
\end{aligned}
$$

where the first step stems from $\sum_h n_{h,T} - n_{h,T}^* = \sum_h n_{h,T} - \sum_h n_{h,T}^* = T - T = 0$, the second step stems from Lemma 3, and the last steps stem from taking the max over the sum. The last inequality implies

$$\|\boldsymbol{n} - \boldsymbol{n}^*\|_\infty \leq 3G + \frac{4GC_T p}{\Sigma_p(p+1)} \left(\min \sigma_g + \frac{2C_T}{\sqrt{\min_h n_{h,T}^*}}\right)^{\frac{p-1}{p+1}} \sqrt{\min_h n_{h,T}^*}. \quad (23)$$

In particular, $\|\boldsymbol{n} - \boldsymbol{n}^*\|_\infty = \tilde{O}(\sqrt{\min_h n_{h,t}^*}) = \tilde{O}(\sqrt{T})$ and

$$\max_g \frac{n_{g,T}^*}{n_{g,T}} \leq \frac{1}{1 - \frac{\|\boldsymbol{n} - \boldsymbol{n}^*\|_\infty}{\min_h n^*}} = \frac{1}{1 - \tilde{O}(T^{-0.5})} = \tilde{O}(1).$$

Therefore,

$$
\begin{aligned}
\frac{p+1}{2} \frac{G\|n - n^*\|_\infty^2}{T\min_g n_{g,T}^*} + \frac{7(p+2)^2\Sigma_p^2}{\sigma_{\min}^2} \max_g \left(\frac{n_{g,T}^*}{n_g}\right)^{3p+3} \frac{\|\boldsymbol{n} - \boldsymbol{n}^*\|_\infty^3}{T^3} &= \frac{p+1}{2} \frac{G\tilde{O}(T)}{T\Theta(T)} + \frac{7(p+2)^2\Sigma_p^2}{\sigma_{\min}^2}\tilde{O}(1)\frac{\tilde{O}(T^{1.5})}{T^3} \\
&= \tilde{O}(T^{-1}).
\end{aligned}
$$

Recall from Lemma 1 that $R_p^* = R_p(\boldsymbol{n}^*) = \Theta(T^{-1})$. Thus by taking the conditional expectation on $\mathcal{A}_T$, we have

$$\mathbb{E}[R_p(\boldsymbol{n}) - R_p^*|\mathcal{A}_T] \leq R_p^* \tilde{O}(T^{-1}) = \tilde{O}(T^{-2}),$$

which concludes the proof of Theorem 1. $\qquad\square$

## A.6 Proof of auxiliary lemmas

In this section, we prove Lemmas 11 and 12. To do so, simple expressions for the derivatives of $r_p$ are needed. These are established in the following lemma:

**Lemma 13.** *For $g, h, i \in [G-1]$, we introduce the following functions in $\mathcal{K}^\circ$:*

$$H_g : \boldsymbol{\lambda} \in \mathcal{K} \to \frac{\sigma_G^{2p}}{(1 - \lambda_1 - \ldots - \lambda_{G-1})^{p+1}} - \frac{\sigma_g^{2p}}{\lambda_g^{p+1}}$$

$$G_{h,g} := \frac{1}{p+1} \frac{\partial}{\partial \lambda_g} H_h,$$

$$I_{g,h,i} := \frac{1}{p+2} \frac{\partial}{\partial \lambda_i} G_{g,h}.$$

*The following holds:*

1. $\nabla r_p = r_p^{1-p} \boldsymbol{H}$.

2. $\mathcal{H}_{g,h} = (1-p) r_p^{1-2p} H_g H_h + (p+1) r_p^{1-p} G_{g,h}$.

3.

$$\frac{\partial^3 r_p}{\partial \lambda_g \partial \lambda_h \partial_i} = (1-p)(1-2p) H_g H_h H_i r_p^{1-3p}$$
$$+ (1-2p)(1+p)(H_g G_{h,i} + H_h G_{i,g} + H_g G_{h,i})$$
$$+ r_p^{1-2p} + (p+1)(p+2) I_{g,h,i} r_p^{1-p}.$$

*Proof.* Fix a $\lambda \in \mathcal{K}^\circ$ and $g, h, i \in [G-1]$.

**Expression of the gradient:** On the one hand, the definition of $r_p$ in (17) implies

$$r_p^p(\boldsymbol{\lambda}) = \left\| \left\{ \frac{\sigma_1^2}{\lambda_1}, \ldots, \frac{\sigma_{G-1}^2}{\lambda_{G-1}}, \frac{\sigma_G^2}{1 - \lambda_1 - \ldots - \lambda_{G-1}} \right\} \right\|_p^p = \frac{\sigma_G^{2p}}{(1 - \lambda_1 - \ldots - \lambda_{G-1})^p} + \sum_{h \leq G-1} \frac{\sigma_h^{2p}}{\lambda_h^p},$$

so that

$$\frac{\partial}{\partial \lambda_g}(r_p^p)(\boldsymbol{\lambda}) = p \left[ \frac{\sigma_G^{2p}}{(1 - \lambda_1 - \ldots - \lambda_{G-1})^{p+1}} - \frac{\sigma_g^{2p}}{\lambda_g^{p+1}} \right] = p H_g(\boldsymbol{\lambda}). \tag{24}$$

On the other hand, from the formula $(f^p)' = p f' f^{p-1}$,

$$\frac{\partial}{\partial \lambda_g}(r_p^p)(\boldsymbol{\lambda}) = p r_p^{p-1}(\lambda) \frac{\partial}{\partial \lambda_g} r_p(\boldsymbol{\lambda}), \tag{25}$$

Combining Equations (24) and (25) yields

$$\frac{\partial}{\partial \lambda_g} r_p^p = p H_g = p r_p^{p-1} \frac{\partial}{\partial \lambda_g} r_p,$$

so that

$$\frac{\partial}{\partial \lambda_g} r_p = r_p^{1-p} H_g. \tag{26}$$

Therefore $\nabla r_p = \left( \frac{\partial}{\partial \lambda_1} r_p, \ldots, \frac{\partial}{\partial \lambda_{G-1}} r_p \right) = r_p^{1-p}(H_1, \ldots, H_{G-1}) = r_p^{1-p} \boldsymbol{H}$, which proves the first Eq. of Lemma 13.

**Expression of the Hessian:** We have

$$
\begin{aligned}
\mathcal{H}_{g,h} &= \frac{\partial^2}{\partial\lambda_g\partial\lambda_h} r_p \\
&= \frac{\partial}{\partial\lambda_g}\left(r_p^{1-p}H_h\right) \\
&= (1-p)r_p^{-p}\left(\frac{\partial}{\partial\lambda_g}r_p\right)H_h + r_p^{1-p}\frac{\partial}{\partial\lambda_g}H_h \\
&= (1-p)r_p^{1-2p}H_gH_h + (p+1)r_p^{1-p}G_{g,h}.
\end{aligned}
$$

where the first equality is due to the definition of the Hessian, the second equality is due to the expression of the gradient from Eq. (26), the third equality applies the product rule to the derivative, and the fourth equality applies the definition of $G_{g,h}$ in Lemma 13. This proves the second Equality of Lemma 13.

**Third derivatives.**

$$
\begin{aligned}
\frac{\partial^3}{\partial\lambda_g\partial\lambda_h\partial_i} r_p &= \frac{\partial}{\partial\lambda_i}\mathcal{H}_{g,h} \\
&= \frac{\partial}{\partial\lambda_i}\left\{(1-p)r_p^{1-2p}H_gH_h + (p+1)r_p^{1-p}G_{g,h}\right\} \\
&= (1-p)\frac{\partial}{\partial\lambda_i}\left\{r_p^{1-2p}H_gH_h\right\} + (p+1)\frac{\partial}{\partial\lambda_i}\left\{r_p^{1-p}G_{g,h}\right\},
\end{aligned}
$$

where the first equality is due to the definition of the Hessian, the second inequality is due to the Hessian expression established previously, and the third equality is due to the linearity of derivation. In a similar fashion to the previous case, we derivate each of the products $r_p^{1-2p}H_gH_h$ and $r_p^{1-p}G_{g,h}$ separately. On the one hand,

$$
\begin{aligned}
\frac{\partial}{\partial\lambda_i}\{r_p^{1-2p}H_gH_h\} &= H_gH_h\frac{\partial}{\partial\lambda_i}\{r_p^{1-2p}\} + r_p^{1-2p}\left(H_g\frac{\partial}{\partial\lambda_i}H_h + H_h\frac{\partial}{\partial\lambda_i}H_g\right) \\
&= H_gH_h(1-2p)r_p^{-2p}r_p^{1-p}H_i + r_p^{1-2p}\left(H_g(p+1)G_{h,i} + H_h(p+1)G_{g,i}\right) \\
&= (1-2p)r_p^{1-3p}H_gH_hH_i + (p+1)r_p^{1-2p}\left(H_gG_{h,i} + H_hG_{g,i}\right),
\end{aligned}
$$

where the first equality is due to the derivation product rule, the second equality is due to the definition of $G_{g,h}$ introduced in Lemma 13, and the third equality is due to a reordering of the terms. On the other hand, by following the exact same steps

$$
\begin{aligned}
\frac{\partial}{\partial\lambda_i}\left\{r_p^{1-p}G_{g,h}\right\} &= (1-p)r_p^{-p}\frac{\partial}{\partial\lambda_i}r_p + r_p^{1-p}\frac{\partial}{\partial\lambda_i}G_{g,h} \\
&= (1-p)r_p^{-p}r_p^{1-p}H_iG_{g,h} + (p+2)r_p^{1-p}I_{g,h,i} \\
&= (1-p)r_p^{1-2p}H_iG_{g,h} + (p+2)r_p^{1-p}I_{g,h,i}.
\end{aligned}
$$

Replacing the previous two expressions in the formula for $\frac{\partial^3}{\partial\lambda_g\partial\lambda_h\partial_i} r_p$ yields

$$
\begin{aligned}
\frac{\partial^3}{\partial\lambda_g\partial\lambda_h\partial_i} r_p &= (1-p)(1-2p)H_gH_hH_ir_p^{1-3p} \\
&\quad + (1-2p)(1+p)\left(H_gG_{h,i} + H_hG_{i,g} + H_gG_{h,i}\right) \\
&\quad + r_p^{1-2p} + (p+1)(p+2)I_{g,h,i}r_p^{1-p},
\end{aligned}
$$

where the symmetry of $G_{g,h} = G_{h,g}$ is used. This derives the third Eq. of Lemma 13 and concludes the proof. $\qquad\square$

We are now ready to establish the proof of Lemma 11.

**Lemma 11.** $\langle\mathcal{H}(\boldsymbol{\lambda}^*)(\boldsymbol{\lambda}-\boldsymbol{\lambda}^*),\boldsymbol{\lambda}-\boldsymbol{\lambda}^*\rangle = (p+1)r_p(\boldsymbol{\lambda}^*)\sum_{g\in[G]}\frac{(\lambda_g-\lambda_g^*)^2}{\lambda_g^*}.$

*Proof.* Setting the value $\boldsymbol{\lambda}^*$ in the value of the Hessian established in Lemma 13 implies that for $g, h \in [G]$:

$$\mathcal{H}_{g,h}(\boldsymbol{\lambda}^*) = (1-p)r_p^{1-2p}H_g H_h(\boldsymbol{\lambda}^*) + (p+1)r_p^{1-p}(\boldsymbol{\lambda}^*)G_{g,h}(\boldsymbol{\lambda}^*)$$
$$= (p+1)r_p^{1-p}(\boldsymbol{\lambda}^*)G_{g,h}(\boldsymbol{\lambda}^*)$$
$$= (p+1)r_p^{1-p}(\boldsymbol{\lambda}^*)\Sigma_p^{p+1}\left(\frac{1}{\lambda_G^*} + \frac{1}{\lambda_g^*}\mathbb{1}(g = h)\right)$$
$$= (p+1)r_p(\boldsymbol{\lambda}^*)\left(\frac{1}{\lambda_G^*} + \frac{1}{\lambda_g^*}\mathbb{1}(g = h)\right)$$

where the first equality stems from the definition of the Hessian, the second equality stems from $\boldsymbol{H}(\boldsymbol{\lambda}^*) = 0$ due to the optimality of $\boldsymbol{\lambda}^*$, the third equality stems from the definition of $G_{g,h}$, and the fourth equality stems from $r_p(\boldsymbol{\lambda}^*) = \Sigma_p^{\frac{1}{p}-1}$. As a consequence,

$$\langle\mathcal{H}(\boldsymbol{\lambda}^*)(\boldsymbol{\lambda} - \boldsymbol{\lambda}^*), \boldsymbol{\lambda} - \boldsymbol{\lambda}^*\rangle = \sum_{g,h\in[G-1]} \mathcal{H}_{g,h}(\boldsymbol{\lambda} - \boldsymbol{\lambda}^*)_g(\boldsymbol{\lambda} - \boldsymbol{\lambda}^*)_h$$

$$= (p+1)r_p(\boldsymbol{\lambda}^*)\sum_{g,h\in[G-1]}\left(\frac{1}{\lambda_G^*} + \frac{1}{\lambda_g^*}\mathbb{1}(g = h)\right)(\boldsymbol{\lambda} - \boldsymbol{\lambda}^*)_g(\boldsymbol{\lambda} - \boldsymbol{\lambda}^*)_h$$

$$= (p+1)r_p(\boldsymbol{\lambda}^*)\left\{\frac{\sum_{g,h\in[G-1]}(\boldsymbol{\lambda} - \boldsymbol{\lambda}^*)_g(\boldsymbol{\lambda} - \boldsymbol{\lambda}^*)_h}{\lambda_G^*} + \sum_{g\in[G-1]}\frac{(\boldsymbol{\lambda} - \boldsymbol{\lambda}^*)_g^2}{\lambda_g^*}\right\}$$

$$= (p+1)r_p(\boldsymbol{\lambda}^*)\left\{\frac{\left(\sum_{g\in[G-1]}(\boldsymbol{\lambda} - \boldsymbol{\lambda}^*)_g\right)^2}{\lambda_G^*} + \sum_{g\in[G-1]}\frac{(\boldsymbol{\lambda} - \boldsymbol{\lambda}^*)_g^2}{\lambda_g^*}\right\}$$

$$= (p+1)r_p(\boldsymbol{\lambda}^*)\left\{\frac{(\boldsymbol{\lambda} - \boldsymbol{\lambda}^*)_G^2}{\lambda_G^*} + \sum_{g\in[G-1]}\frac{(\boldsymbol{\lambda} - \boldsymbol{\lambda}^*)_g^2}{\lambda_g^*}\right\}$$

$$= (p+1)r_p(\boldsymbol{\lambda}^*)\sum_{g\in[G]}\frac{(\lambda_g - \lambda_g^*)^2}{\lambda_g^*},$$

where the first step follows from the definition of the scalar product, and the second step stems from the expression of $\mathcal{H}(\boldsymbol{\lambda}^*)$ derived previously in the proof. In the third step, the sum is distributed over the terms $\frac{1}{\lambda_G^*}$ and $\frac{1}{\lambda_g^*}\mathbb{1}(g = h)$. In the fourth step, the first sum is factorized, and in the fifth step, the equality $\sum_{g\in[G-1]}(\boldsymbol{\lambda} - \boldsymbol{\lambda}^*)_g = (1 - \lambda_G) - (1 - \lambda_G^*) = \lambda_G^* - \lambda_G$ is used. This completes the proof of Lemma 11. $\qquad\square$

Next, we establish the proof of Lemma 12.

**Lemma 12.** *The following inequality holds:*

$$\sup_{\substack{\boldsymbol{u}\in[\boldsymbol{\lambda},\boldsymbol{\lambda}'] \\ x,y,z\in\mathbb{N} \\ x+y+z=3 \\ \{g,h,i\}\subset[G-1]}}\left|\frac{1}{x!y!z!}\frac{\partial^3 r_p(\boldsymbol{u})}{\partial\lambda_g\partial\lambda_h\partial\lambda_i}\right| \leq \frac{7(p+2)^2}{(\min_g \lambda_g^*)^2}\max_g\left(\frac{\lambda_g^*}{\lambda_g}\right)^{3p+3}r_p(\boldsymbol{\lambda}^*).$$

*Proof.* From Lemma 1, the following equality holds for each $g \in [G]$:

$$\sigma_g^{2p} = \left(\sum_{h\in[G]}\sigma_h^{\frac{2p}{p+1}}\right)^{p+1}(\lambda_g^*)^{p+1} = \Sigma_p^{p+1}(\lambda_g^*)^{p+1}.$$

For a fixed $\boldsymbol{u} \in [\boldsymbol{\lambda}, \boldsymbol{\lambda}^*]$, $\sum_g u_g = \sum_g \lambda_g = \sum_g \lambda_g^* = 1$, thus the coordinate $g_0$ achieving the maximal $\frac{\lambda_g^*}{u_g}$ must have $u_{g_0} \leq \lambda_{g_0}^*$. Since $\boldsymbol{u} \in [\boldsymbol{\lambda}, \boldsymbol{\lambda}^*]$ this implies that $\lambda_{g_0} \leq u_{g_0} \leq \lambda_{g_0}^*$ and

consequently $\max_g \frac{\lambda_g^*}{u_g} \le \max_g \frac{\lambda_g^*}{\lambda_g}$. The following upper bounds follow:

$$|H_g(\boldsymbol{u})| = \Sigma_p^{p+1} \left| \left(\frac{\lambda_G^*}{u_G}\right)^{p+1} - \left(\frac{\lambda_g^*}{u_g}\right)^{p+1} \right| \qquad \le \Sigma_p^{p+1} \left( \max_g \frac{\lambda_g^*}{\lambda_g} \right)^{p+1},$$

$$|G_{g,h}(\boldsymbol{u})| = \Sigma_p^{p+1} \left| \frac{(\lambda_G^*)^{p+1}}{(u_G)^{p+2}} + \frac{(\lambda_g^*)^{p+1}}{(u_g)^{p+2}} \mathbb{1}(g=h) \right| \qquad \le \frac{2\Sigma_p^{p+1}}{\min_g \lambda_g^*} \left( \max_g \frac{\lambda_g^*}{\lambda_g} \right)^{p+2,}$$

$$|I_{g,h,i}(\boldsymbol{u})| = \Sigma_p^{p+1} \left| \frac{(\lambda_G^*)^{p+1}}{(u_G)^{p+3}} - \frac{(\lambda_g^*)^{p+1}}{(u_g)^{p+3}} \mathbb{1}(g=h=i) \right| \quad \le \frac{\Sigma_p^{p+1}}{(\min_g \lambda_g^*)^2} \left( \max_g \frac{\lambda_g^*}{\lambda_g} \right)^{p+3}.$$

Moreover, since $p \ge 1$, each $j \in \{1-p, 1-2p, 1-3p\}$ is non-positive, and by minimality of $\boldsymbol{\lambda}^*$:

$$r_p(\boldsymbol{u})^j \le r_p(\boldsymbol{\lambda}^*)^j,$$

so that:

$$|H_g H_h H_i r_p^{1-3p}(\boldsymbol{u})| \le \left( \max_g \frac{\lambda_g^*}{\lambda_g} \right)^{3p+3} \Sigma_p^{3p+3} r_p(\boldsymbol{\lambda}^*)^{1-3p},$$

$$\left| \left( H_g G_{h,i} + H_h G_{i,g} + H_g G_{h,i} \right) r_p^{1-2p}(\boldsymbol{u}) \right| \le 3 \frac{2\Sigma_p^{p+1}}{\min_g \lambda_g^*} \left( \max_g \frac{\lambda_g^*}{\lambda_g} \right)^{p+2} \left( \Sigma_p^{p+1} \left( \max_g \frac{\lambda_g^*}{\lambda_g} \right)^{p+1} \right),$$

$$|I_{g,h,i} r_p^{1-p}(\boldsymbol{u})| \le \frac{\Sigma_p^{p+1}}{(\min_g \lambda_g^*)^2} \left( \max_g \frac{\lambda_g^*}{\lambda_g} \right)^{p+3} r_p^{1-p}(\boldsymbol{\lambda}^*).$$

Each of the previous three expressions can be simplified by using $r_p(\boldsymbol{\lambda}^*) = \Sigma_p^{\frac{1}{p}-1}$:

$$|H_g H_h H_i r_p^{1-3p}(\boldsymbol{u})| \le \left( \max_g \frac{\lambda_g^*}{\lambda_g} \right)^{3p+3} r_p(\boldsymbol{\lambda}^*),$$

$$\left| \left( H_g G_{h,i} + H_h G_{i,g} + H_g G_{h,i} \right) r_p^{1-2p}(\boldsymbol{u}) \right| \le \frac{6}{\min_g \lambda_g^*} \left( \max_g \frac{\lambda_g^*}{\lambda_g} \right)^{2p+3} r_p(\boldsymbol{\lambda}^*),$$

$$|I_{g,h,i} r_p^{1-p}(\boldsymbol{u})| \le \frac{1}{(\min_g \lambda_g^*)^2} \left( \max_g \frac{\lambda_g^*}{\lambda_g} \right)^{p+3} r_p(\boldsymbol{\lambda}^*).$$

Hence by using the expression of the third derivatives established in Lemma 13:

$$\left| \frac{\partial^3 r_p(\boldsymbol{u})}{\partial \lambda_g \partial \lambda_h \partial \lambda_i} \right| \le \left| (1-p)(1-2p) H_g H_h H_i r_p^{1-3p}(\boldsymbol{u}) \right|$$

$$+ \left| (1-2p)(1+p) \left( H_g G_{h,i} + H_h G_{i,g} + H_g G_{h,i} \right) r_p^{1-2p(\boldsymbol{u})} \right|$$

$$+ \left| (p+1)(p+2) I_{g,h,i} r_p^{1-p}(\boldsymbol{u}) \right|$$

$$\le 2p^2 \left( \max_g \frac{\lambda_g^*}{\lambda_g} \right)^{3p+3} r_p(\boldsymbol{\lambda}^*) + \frac{12(p+1)^2}{\min_g \lambda_g^*} \left( \max_g \frac{\lambda_g^*}{\lambda_g} \right)^{2p+3} r_p(\boldsymbol{\lambda}^*)$$

$$+ \frac{(p+2)^2}{(\min_g \lambda_g^*)^2} \left( \max_g \frac{\lambda_g^*}{\lambda_g} \right)^{p+3} r_p(\boldsymbol{\lambda}^*)$$

$$\le \frac{7(p+2)^2}{(\min_g \lambda_g^*)^2} \left( \max_g \frac{\lambda_g^*}{\lambda_g} \right)^{3p+3} r_p(\boldsymbol{\lambda}^*).$$

As a consequence, by taking the sup over $u, g, h, i, x, y, z$,

$$\sup_{\substack{u \in [\lambda, \lambda^*] \\ x,y,z \in \mathbb{N} \\ x+y+z=3 \\ \{g,h,i\} \subset [G-1]}} \left| \frac{1}{x! y! z!} \frac{\partial^3 r_p(\boldsymbol{u})}{\partial \lambda_g \partial \lambda_h \partial \lambda_i} \right| \le \frac{7(p+2)^2}{(\min_g \lambda_g^*)^2} \max_g \left( \frac{\lambda_g^*}{\lambda_g} \right)^{3p+3} r_p(\boldsymbol{\lambda}^*), \qquad (27)$$

which completes the proof of Lemma 12. □

# B Proof of Theorem 2

In this section, we prove Theorem 2. First, we establish an initial lower bound that captures the trade-off between how hard it is to distinguish two instances and how hard it is to optimize both under the same action (See Appendix B.1). Next, we provide a specific counter example which regret is at least $\Theta(T^{-2})$ (See Appendix B.2).

## B.1 Proof of Lemma 6

**Lemma 6.** *Let $\boldsymbol{\pi}$ be a fixed policy and $\mathcal{D}^a$, $\mathcal{D}^b$ be two instances with standard deviation vectors $\boldsymbol{\sigma}^a, \boldsymbol{\sigma}^b$, respectively. Then,*

$$\max\{Regret_{p,T}(\boldsymbol{\pi}, \mathcal{D}^a), Regret_{p,T}(\boldsymbol{\pi}, \mathcal{D}^b)\} \geq d(\boldsymbol{\sigma}^a, \boldsymbol{\sigma}^b) \exp\left(-\sum_{g \in [G]} \mathbb{E}_{\boldsymbol{\pi}, \mathcal{D}^a}[n_{g,t}] KL(\mathcal{D}_g^a || \mathcal{D}_g^b)\right).$$

*Proof.* Let $\mathcal{D}^a$ and $\mathcal{D}^b$ be two instances with standard deviations $\boldsymbol{\sigma}^a$ and $\boldsymbol{\sigma}^b$, and let $\delta \geq d(\sigma^a, \sigma^b)$. Let $X$ be a random variable over the set $\{a, b\}$. The following inequalities hold

$$\begin{aligned}
\max(\text{Regret}_{p,T}(\boldsymbol{\pi}, \mathcal{D}^a), \text{Regret}_{p,T}(\boldsymbol{\pi}, \mathcal{D}^b)) &\geq \mathbb{E}_X[\text{Regret}_p(\boldsymbol{\pi}, \mathcal{D}^X)] \\
&\geq \mathbb{E}[R_p(\boldsymbol{n}; \boldsymbol{\sigma}^X) - R_p^*(\boldsymbol{\sigma}^X) | R_p(\boldsymbol{n}; \boldsymbol{\sigma}^X) - R_p^*(\boldsymbol{\sigma}^X) > \delta] \\
&\quad \times \mathbb{P}_{\boldsymbol{\pi}, X}\left(R_p(\boldsymbol{n}; \boldsymbol{\sigma}^X) - R_p^*(\boldsymbol{\sigma}^X) > \delta\right) \\
&\geq \delta \mathbb{P}_{X, \boldsymbol{\pi}}\left(R_p(\boldsymbol{n}; \boldsymbol{\sigma}^X) - R_p^*(\boldsymbol{\sigma}^X) > \delta\right),
\end{aligned}$$

where the first step stems from the support of $X$ being $\{a, b\}$, and the second step stems from the law of total expectation. Let $\hat{x}$ be the following (random) classifier:

$$\hat{x} := \begin{cases} a & \text{If } R_p(\boldsymbol{n}; \boldsymbol{\sigma}^a) - R_p^*(\boldsymbol{\sigma}^a) \leq \delta \\ b & \text{If } R_p(\boldsymbol{n}, \boldsymbol{\sigma}^b) - R_p^*(\boldsymbol{\sigma}^b) \leq \delta \\ \text{Indifferent} & \text{Otherwise} \end{cases}$$

Since $\delta \geq d(\sigma^a, \sigma^b)$, $\hat{x}$ is well defined. Moreover, $\mathbb{P}_{\boldsymbol{\pi}.X}\left(R_p(\boldsymbol{n}; \boldsymbol{\sigma}^X) - R_p^*(\boldsymbol{\sigma}^X) > \delta\right) \geq \mathbb{P}_X(\hat{x} \neq X) \geq \inf_{\hat{x}} \mathbb{P}_X(\hat{x} \neq X)$, where the infimum is taken over all the classifiers of $\{a, b\}$. Moreover, by Pinsker's inequality, $\inf_{\hat{x}} \mathbb{P}_X(\hat{x} \neq X) \geq \exp\left(-KL(\mathcal{D}^a || \mathcal{D}^b)\right)$. Moreover, from bandits feedback divergence properties (see [25]), $KL(\mathcal{D}^a || \mathcal{D}^b) = \sum_{g \in [G]} \mathbb{E}_{\boldsymbol{\pi}, \mathcal{D}^a}[n_{g,t}] KL(\mathcal{D}_g^a || \mathcal{D}_g^b)$, which concludes the proof of Lemma 6. $\qquad\square$

## B.2 The counter-examples

We introduce the following instances for each $g \in [G]$,

$$\mathcal{D}^g : \begin{cases} \mathcal{D}_1^a \sim \mathcal{N}\left(0, 1 + \frac{1}{\sqrt{T}}\right) \\ \mathcal{D}_g^a \sim \mathcal{N}(0, 1), & \forall g \neq 1 \end{cases} \qquad \mathcal{D}^b : \begin{cases} \mathcal{D}_2^b \sim \mathcal{N}\left(0, 1 + \frac{1}{\sqrt{T}}\right) \\ \mathcal{D}_g^a \sim \mathcal{N}(0, 1), & \forall g \neq 2 \end{cases}$$

We start by upper bounding the $KL$-divergence between the two instances:

**Lemma 14.** *The following inequality holds:* $\sum_{g \in [G]} \mathbb{E}_{\boldsymbol{\pi}, \mathcal{D}^a}[n_{g,T}] KL(\mathcal{D}_g^a || \mathcal{D}_g^b) \leq \frac{1}{2}$

*Proof.* For convenience, we set $\nu := \sqrt{\frac{1}{T}} < 1$. The formula for the $KL-$divergence of two univariate normal distributions of zero mean implies

$$KL(\mathcal{D}_h^a || \mathcal{D}_h^b) = \frac{1}{2}\left(\log\left(\frac{\sigma^2(1+\nu)}{\sigma^2}\right) + \frac{\sigma^2 - (\sigma^2(1+\nu))}{\sigma^2(1+\nu)}\right).$$

The taylor expansion of the expression above can be derived by combining the expansions of both the functions $x \to \log(1+x)$ and $x \to \frac{1}{1+x}$ in the domain $(0,1)$:

$$
\frac{1}{2}\left(\log\left(\frac{\sigma^2(1+\nu)}{\sigma^2}\right) + \frac{\sigma^2 - (\sigma^2(1+\nu))}{\sigma^2(1+\nu)}\right) = \frac{1}{2}\left(-\sum_{k\geq 1}\frac{(-1)^k}{k}\nu^k - \nu\sum_{k\geq 0}(-1)^k\nu^k\right)
$$

$$
= \frac{1}{2}\sum_{k\geq 1}(-1)^k\nu^k\left(1 - \frac{1}{k}\right)
$$

$$
= \frac{\nu^2}{2}\sum_{k\geq 0}(-1)^k\nu^k\left(1 - \frac{1}{k+2}\right)
$$

$$
\leq \frac{\nu^2}{2},
$$

Summing over all coordinates implies

$$
\sum_{g\in[G]}\mathbb{E}_{\boldsymbol{\pi},\boldsymbol{\mathcal{D}}^a}[n_{g,T}]KL(\mathcal{D}_g^a||\mathcal{D}_g^b) = \mathbb{E}_{\boldsymbol{\pi},\boldsymbol{\mathcal{D}}^a}[n_{h,T}]KL(\mathcal{D}_h^a||\mathcal{D}_h^b) \leq \frac{T\nu^2}{2} = \frac{1}{2},
$$

where the inequality follows from $\boldsymbol{n}_T \leq T$. This concludes the proof. $\qquad\square$

Next, we derive a simpler form for $d(\boldsymbol{\sigma}^a, \boldsymbol{\sigma}^b)$. The simplification exploits the symmetries in $\boldsymbol{\sigma}^a, \boldsymbol{\sigma}^b$:

**Lemma 15.** *Let $\boldsymbol{u}$ denote the unit vector $(1,\ldots,1)^T$. The following equality holds:*

$$
d(\boldsymbol{\sigma}^a, \boldsymbol{\sigma}^b) = r_p\left(\frac{1}{G}\boldsymbol{u}; \sigma^a\right) - r_p^*(\boldsymbol{\sigma}^a)
$$

*where the function $r_p$ is introduced in Appendix A.*

*Proof.* For $x \in \{a,b\}$, let $\mathcal{S}_\epsilon^x := \{\epsilon > 0 | r_p(\boldsymbol{\lambda}; \sigma^x) - r_p^*(\boldsymbol{\sigma}^x) \leq \epsilon\}$. By definition of $d$,

$$
d(\boldsymbol{\sigma}^a, \boldsymbol{\sigma}^b) = \inf\{\delta \geq 0 | \mathcal{S}_\epsilon^a \cap \mathcal{S}_\epsilon^b \neq \emptyset\}.
$$

We prove $d(\boldsymbol{\sigma}^a, \boldsymbol{\sigma}^b) = r_p\left(\frac{1}{G}\boldsymbol{u}; \sigma^a\right) - r_p^*(\boldsymbol{\sigma}^a)$ by proving each of the inequalities $d(\boldsymbol{\sigma}^a, \boldsymbol{\sigma}^b) \leq r_p\left(\frac{1}{G}\boldsymbol{u}; \sigma^a\right) - r_p^*(\boldsymbol{\sigma}^a)$ and $d(\boldsymbol{\sigma}^a, \boldsymbol{\sigma}^b) \geq r_p\left(\frac{1}{G}\boldsymbol{u}; \sigma^a\right) - r_p^*(\boldsymbol{\sigma}^a)$.

First, we prove $d(\boldsymbol{\sigma}^a, \boldsymbol{\sigma}^b) \leq r_p\left(\frac{1}{G}\boldsymbol{u}; \sigma^a\right) - r_p^*(\boldsymbol{\sigma}^a)$. Since $\boldsymbol{\sigma}^a$ can be obtained by swapping the first two coordinates of $\boldsymbol{\sigma}^b$, the symmetry of $r_p$ implies $r_p^*(\boldsymbol{\sigma}^a) = r_p^*(\boldsymbol{\sigma}^b)$. Moreover, for each $(\lambda_1, \lambda_2, \boldsymbol{\lambda}') \in \mathcal{K}$, $r_p((\lambda_1, \lambda_2, \boldsymbol{\lambda}_{3:G}); \boldsymbol{\sigma}^a) = r_p((\lambda_2, \lambda_1, \boldsymbol{\lambda}_{3:G}); \boldsymbol{\sigma}^b)$. As a consequence, $r_p\left(\frac{1}{G}\boldsymbol{u}; \sigma^a\right) - r_p^*(\boldsymbol{\sigma}^a) = r_p\left(\frac{1}{G}\boldsymbol{u}; \sigma^b\right) - r_p^*(\boldsymbol{\sigma}^b)$, and any $\epsilon > 0$ satisfying $\epsilon \geq r_p\left(\frac{1}{G}\boldsymbol{u}; \sigma^a\right) - r_p^*(\boldsymbol{\sigma}^a)$ must also satisfy $\mathcal{S}_\epsilon^a \cap \mathcal{S}_\epsilon^b \neq \emptyset$. In particular, $d(\boldsymbol{\sigma}^a, \boldsymbol{\sigma}^b) \leq r_p\left(\frac{1}{G}\boldsymbol{u}; \sigma^a\right) - r_p^*(\boldsymbol{\sigma}^a)$.

To derive $d(\boldsymbol{\sigma}^a, \boldsymbol{\sigma}^b) \geq r_p\left(\frac{1}{G}\boldsymbol{u}; \sigma^a\right) - r_p^*(\boldsymbol{\sigma}^a)$, we use the following lemma, which proof is deferred later in the section:

**Lemma 16.** *If $\boldsymbol{\lambda} = (\lambda_1, \ldots, \lambda_G) \in \mathcal{S}_\epsilon^a \cap \mathcal{S}_\epsilon^b$, and $\tau$ a permutation of $[G]$. $\boldsymbol{\lambda}_\tau := (\lambda_{\tau(1)}, \ldots, \lambda_{\tau(G)})$ is also in $\mathcal{S}_\epsilon^a \cap \mathcal{S}_\epsilon^b$.*

Let $\epsilon \geq 0$ satisfying $\mathcal{S}_\epsilon^a \cap \mathcal{S}_\epsilon^b \neq \emptyset$, and let $\boldsymbol{\lambda} \in \mathcal{S}_\epsilon^a \cap \mathcal{S}_\epsilon^b$. For each permutation $\tau$, $\boldsymbol{\lambda}_\tau$ is also in $\mathcal{S}_\epsilon^a \cap \mathcal{S}_\epsilon^b$. Each of $\mathcal{S}_\epsilon^a$ and $\mathcal{S}_\epsilon^b$ is convex and therefore the intersection $\mathcal{S}_\epsilon^a \cap \mathcal{S}_\epsilon^b$ is also convex, which implies that

$$
\frac{1}{G}\boldsymbol{u} = \frac{1}{G!}\sum_{\tau\,\text{permutation}}\boldsymbol{\lambda}_\tau \in \mathcal{S}_\epsilon^a \cap \mathcal{S}_\epsilon^b,
$$

which in turn implies that $\epsilon \geq r_p\left(\frac{1}{G}\boldsymbol{u}, \sigma^a\right) - r^*(\sigma^a)$. By taking the inf we get $d_p(\boldsymbol{\sigma}^a, \boldsymbol{\sigma}^b) \geq r_p\left(\frac{1}{G}\boldsymbol{u}, \sigma^a\right) - r_p^*(\boldsymbol{\sigma}^a)$, which completes the proof. $\qquad\square$

We now state the proof of Lemma 16:

*Proof.* Let $\boldsymbol{\lambda} = (\lambda_1, \ldots, \lambda_G) \in \mathcal{S}_\epsilon^a \cap \mathcal{S}_\epsilon^b$. Since every permutation can be written as a composition of transpositions (2-cycles), it suffices to prove the result for transpositions. Let $\tau = (g, h)$ with $g \neq h$. We distinguish 3 cases:

1. $g, h \geq 3$: Since $\boldsymbol{\sigma}_{3:G}^a = \boldsymbol{\sigma}_{3:G}^b$, $r_p(\boldsymbol{\lambda}; \boldsymbol{\sigma}^x) = r_p(\boldsymbol{\lambda}_\tau; \boldsymbol{\sigma}^x)$ for each $x \in \{a, b\}$. The implication $\boldsymbol{\lambda}_\tau \in \mathcal{S}_\epsilon^a \cap \mathcal{S}_\epsilon^b$ follows.

2. $g, h \leq 2$: By symmetry of the problem, $\boldsymbol{\sigma}_1^a = \boldsymbol{\sigma}_2^b = \boldsymbol{\sigma}_{\tau(1)}^b$. Similarly, $\boldsymbol{\sigma}_2^a = \boldsymbol{\sigma}_{\tau(2)}^b$. Therefore, $\boldsymbol{\lambda} \in \mathcal{S}_\epsilon^a$ implies $\boldsymbol{\lambda}_\tau \in \mathcal{S}_\epsilon^b$. Similarly, $\boldsymbol{\lambda} \in \mathcal{S}_\epsilon^b$ implies $\boldsymbol{\lambda}_\tau \in \mathcal{S}_\epsilon^a$. The implication $\boldsymbol{\lambda}_\tau \in \mathcal{S}_\epsilon^a \cap \mathcal{S}_\epsilon^b$ follows.

3. $g \in \{1, 2\}$ and $h \geq 3$. Without loss of generality, assume that $g = 1$ and $h = 3$. Since $\mathcal{D}_1^b \sim \mathcal{D}_3^b$, the equality $\sigma_1^b = \sigma_3^b$ holds and therefore $r_p(\boldsymbol{\lambda}; \sigma^b) = r_p(\boldsymbol{\lambda}_\tau; \sigma^b)$, therefore $\boldsymbol{\lambda}_\tau \in \mathcal{S}_\epsilon^b$. Moreover, $r_p(\boldsymbol{\lambda}_\tau; \boldsymbol{\sigma}^a) = r_p(\boldsymbol{\lambda}; \sigma^b)$, therefore $\boldsymbol{\lambda}_\tau \in \mathcal{S}_\epsilon^a$. The implication $\boldsymbol{\lambda}_\tau \in \mathcal{S}_\epsilon^a \cap \mathcal{S}_\epsilon^b$ follows.

The three cases cover all possible 2-cycles, which completes the proof. $\qquad \square$

We are now ready to state the proof of Theorem 2. It remains to show that $d_p(\boldsymbol{\sigma}^a, \boldsymbol{\sigma}^b) = r_p \left( \frac{1}{2}\mathbf{u}, \sigma^a \right) - r_p^*(\sigma^a) = \Theta(T^{-2})$. We are ready to state the proof of Theorem 2.

**Theorem 2.** *Let $p$ be finite and $\kappa$ be a universal constant. For any online policy $\boldsymbol{\pi}$, there exists an instance $\mathcal{D}_{\boldsymbol{\pi}}$ such that for any $T \geq 1$,*

$$Regret_{p,T}(\boldsymbol{\pi}, \mathcal{D}_{\boldsymbol{\pi}}) \geq \kappa(p+1)T^{-2} + O\left(T^{-2.5}\right) = \Theta(T^{-2}).$$

*Proof.* We set $\mathcal{D}_{\boldsymbol{\pi}} := \operatorname{argmax} \left\{ \text{Regret}_p\left(\boldsymbol{\pi}, \mathcal{D}^a\right), \text{Regret}_p\left(\boldsymbol{\pi}, \mathcal{D}^b\right) \right\}$. The following inequalities hold:

$$\text{Regret}_p\left(\boldsymbol{\pi}, \mathcal{D}_{\boldsymbol{\pi}}\right) \geq \max\left\{ \text{Regret}_p\left(\boldsymbol{\pi}, \mathcal{D}^a\right), \text{Regret}_p\left(\boldsymbol{\pi}, \mathcal{D}^b\right) \right\}$$

$$\geq d(\boldsymbol{\sigma}^a, \boldsymbol{\sigma}^b) \exp\left( -\sum_{g \in [G]} \mathbb{E}_{\boldsymbol{\pi}, \mathcal{D}^a}[n_{g,t}] KL(\mathcal{D}_g^a || \mathcal{D}_g^b) \right)$$

$$\geq \left( r_p\left(\frac{1}{G}\boldsymbol{u}; \sigma^a\right) - r_p^*(\boldsymbol{\sigma}^a) \right) \exp\left(\frac{-1}{2}\right)$$

where the first step stems from the definition of $\mathcal{D}_{\boldsymbol{\pi}}$, the second step stems from Lemma 6, and the third step stems from a combination of both Lemma 19 and Lemma 15. The rest of the proof consists in deriving a lower bound on $r_p\left(\frac{1}{G}\boldsymbol{u}; \sigma^a\right) - r_p^*(\boldsymbol{\sigma}^a)$. Since we will now solely focus on $\boldsymbol{\sigma}^a$, we can drop the dependencies in $\boldsymbol{\sigma}$. A direct consequence of Inequality (19) is

$$\frac{r_p\left(\frac{1}{G}\boldsymbol{u}\right)}{r_p^*} - 1 \geq \frac{p+1}{2} \sum_{g \in [G]} \frac{\left(\frac{1}{G} - \lambda_g^*\right)^2}{\lambda_g^*} - \frac{7(p+2)^2}{(\min_g \lambda_g^*)^2} \max_g \left(\frac{\lambda_g^*}{1/G}\right)^{3p+3} \|\boldsymbol{u}/G - \boldsymbol{\lambda}^*\|_\infty^3. \quad (28)$$

We will lower bound each of the terms $\frac{p+1}{2} \sum_{g \in [G]} \frac{\left(\frac{1}{G} - \lambda_g^*\right)^2}{\lambda_g^*}$ and $-\frac{7(p+2)^2}{(\min_g \lambda_g^*)^2} \max_g \left(\frac{\lambda_g^*}{1/G}\right)^{3p+3} \|\boldsymbol{u}/G - \boldsymbol{\lambda}^*\|_\infty^3$. For convenience, we set

$$f_T := \lambda_1^* - \frac{1}{G}.$$

Since the first group has the highest variance, $\lambda_1^*$ should also be the highest. In particular, $\lambda_1^* \geq \frac{1}{G}$ and $f_T \geq 0$. By symmetry of $\boldsymbol{\sigma}^a$, $\lambda_2^* = \ldots = \lambda_G^* = \frac{1}{G} - \frac{f_T}{G-1}$. Therefore,

$$\sum_{g \in [G]} \frac{\left(\frac{1}{G} - \lambda_g^*\right)^2}{\lambda_g^*} = \frac{f_T^2}{\frac{1}{G} + f_T} + \frac{(G-1)\left(\frac{f_T}{G-1}\right)^2}{\frac{1}{G} - \frac{f_T}{G-1}} = Gf_T^2 \left( \underbrace{\frac{1}{1 + Gf_T}}_{\geq 1 - Gf_T} + \underbrace{\frac{1}{G-1}\frac{1}{1 - \frac{Gf_T}{G-1}}}_{\geq 0} \right) \geq Gf_T^2 - G^2 f_T^3.$$

Next, notice that $\boldsymbol{\lambda}^* \leq 1$, which implies that $\max_g \left(\frac{\lambda_g^*}{1/G}\right)^{3p+3} \leq G^{3p+3}$, and that $\|\boldsymbol{u}/G - \boldsymbol{\lambda}^*\|_\infty = \max(f_T, f_T/(G-1)) = f_T$, and $\min_g \lambda_g^* = \frac{1}{G} - \frac{f_T}{G-1}$. Therefore,

$$
\frac{r_p\left(\frac{1}{G}\boldsymbol{u}\right)}{r_p^*} - 1 \geq \frac{p+1}{2}\left(Gf_T^2 - G^2 f_T^3\right) - \frac{7(p+2)^2 G^{3p+3} f_T^3}{\left(\frac{1}{G} - \frac{f_T}{G-1}\right)^2} = \frac{p+1}{2}\left(Gf_T^2 - G^2 f_T^3\right) - \frac{7(p+2)^2 G^{3p+5} f_T^3}{\left(1 - \frac{G}{G-1}f_T\right)^2}.
$$
$$(29)$$

It remains to bound $f_T$. We do so by deriving the first terms of its taylor expansion in and $T$:

$$
\begin{aligned}
f_T &= \frac{\left(1 + \frac{1}{\sqrt{T}}\right)^{\frac{2p}{p+1}}}{G - 1 + \left(1 + \frac{1}{\sqrt{T}}\right)^{\frac{2p}{p+1}}} - \frac{1}{G} \\
&= \frac{1 + \frac{2p}{p+1}\frac{1}{\sqrt{T}} + o\left(\frac{1}{\sqrt{T}}\right)}{G + \frac{2p}{p+1}\frac{1}{\sqrt{T}} + o\left(\frac{1}{\sqrt{T}}\right)} - \frac{1}{G} \\
&= \frac{1}{G}\left(\left(1 + \frac{2p}{p+1}\frac{1}{\sqrt{T}} + o\left(\frac{1}{\sqrt{T}}\right)\right)\left(1 - \frac{2p}{p+1}\frac{1}{G\sqrt{T}} + o\left(\frac{1}{\sqrt{T}}\right)\right) - 1\right) \\
&= \frac{2p}{p+1}\frac{1}{G}\left(1 - \frac{1}{G}\right)\frac{1}{\sqrt{T}} + o\left(\frac{1}{\sqrt{T}}\right),
\end{aligned}
$$

where the first equality stems from Lemma 1, the second equality stems from the binomial Taylor expansion, and the third equality stems from the Taylor expansion of $x \to \frac{1}{1-x}$. The last step implies that there exists universal constants $q_1, q_2$ such that for $T \geq 1$,

$$
\frac{1}{G}\left(1 - \frac{1}{G}\right)\frac{q_1}{\sqrt{T}} \leq f_T \leq \frac{1}{G}\left(1 - \frac{1}{G}\right)\frac{q_2}{\sqrt{T}}.
$$

Replacing the previous bounding of $f_T$ in Inequality (29) yields

$$
\frac{r_p\left(\frac{1}{G}\boldsymbol{u}\right)}{r_p^*} - 1 \geq \frac{2q_1^2}{G}\left(1 - \frac{1}{G}\right)^2\frac{p+1}{T} + O\left(\frac{1}{T\sqrt{T}}\right),
$$

which in turn implies

$$
r_p\left(\frac{1}{G}\boldsymbol{u}\right) - r_p^* \geq \frac{q_1^2(p+1)}{4}\frac{r_p^*}{GT} + O\left(\frac{1}{T^2\sqrt{T}}\right) \geq \frac{q_1^2}{4}\frac{p+1}{T^2} + O\left(\frac{1}{T^2\sqrt{T}}\right).
$$

where we used $r_p^* = \Theta(T^{-1})$ and from Lemma 1 that

$$
r_p^* = \left\|\frac{\boldsymbol{\sigma}^2}{\boldsymbol{n}_T^*}\right\|_p = \frac{1}{T}\Sigma_p^{\frac{p+1}{p}} = \frac{\left((G-1) + \left(1 + \frac{1}{\sqrt{T}}\right)^{\frac{2p}{p+1}}\right)^{\frac{p+1}{p}}}{T} \geq \frac{G}{T}.
$$

By setting $\kappa = \frac{q_1^2}{4}$, we get $\text{Regret}_T(\boldsymbol{\pi}, \mathcal{D}_{\boldsymbol{\pi}}) \geq \kappa\frac{p+1}{T^2} + O\left(\frac{1}{T^2\sqrt{T}}\right)$, which completes the proof of Theorem 4. $\qquad\square$

## C  Upper and lower bounds when $p = \infty$

The proof for Theorem 3 (upper bound when $p = \infty$) follows the same high-level steps as the proof of Theorem 1 (upper bound when $p \in \mathbb{R}$). However, some adjustments of the proofs are necessary. Table 2 summarizes the changes that are required.

In Appendix C.1, we introduce and prove Lemma 5, the replacement for Lemma 4. In Appendix C.3, we prove Theorem 4.

| Result | Does it hold for $p = +\infty$ ? |
|--------|----------------------------------|
| Lemma 1 | Yes |
| Lemmas 2, 3 | Yes |
| Lemma 4 | No, replaced by Lemma 5 |
| Lemmas 6, 19, 15 | Yes |
| $d(\sigma^a, \sigma^b) = \Theta(T^{-2})$ | No, replaced by $d(\sigma^a, \sigma^b) = \Theta(T^{-1.5})$ |

Table 2: Summary of the possible extensions to $p = +\infty$

## C.1 Curvature of $R_\infty$

The upper bound Lemma 4 goes to $+\infty$ as $p = +\infty$ and is no longer insightful. In this section, we provide a suitable bound:

**Lemma 5.** *Let $\boldsymbol{\sigma} \in \mathbb{R}_+^G$ and $\boldsymbol{n}' \in \mathbb{R}_+^G$ such that $\sum_{g \in [G]} n'_g = T$. Then,*

$$\frac{R_\infty(\boldsymbol{n}'_T) - R_\infty(\boldsymbol{n}^*_T)}{R_\infty(\boldsymbol{n}^*_T)} \leq -\min_g \left( \frac{n'_g}{n^*_{g,T}} - 1 \right) + \frac{1}{4} \max_g \left( \frac{n'_g}{n^*_{g,T}} - 1 \right)^2 \max_g \left( \frac{n^*_{g,T}}{n'_g} \right)^3$$

*Proof.* $\qquad\qquad\qquad\qquad\qquad\qquad\qquad\qquad\qquad\qquad\qquad\qquad\qquad\qquad\qquad\qquad$ $\square$

From Lemma 1,

$$\frac{\sigma_1^2}{n^*_{1,T}} = \ldots = \frac{\sigma_G^2}{n^*_{G,T}} = R_\infty(\boldsymbol{n}^*_T) = R^*_\infty,$$

so that for $g \in [G]$ and $\boldsymbol{n}'$,

$$\frac{\sigma_g^2}{n'_g} = R^*_\infty + \sigma_g^2 \left( \frac{1}{n'_g} - \frac{1}{n^*_{g,T}} \right)$$

$$\leq R^*_\infty + \sigma_g^2 \left( \frac{-1}{(n^*_{g,T})^2}(n'_g - n^*_{g,T}) + \frac{1}{2}(n'_g - n^*_{g,T})^2 \sup_{x \in \{n'_g, n^*_{g,T}\}} \frac{1}{2x^3} \right)$$

$$= R^*_\infty \left( 1 - \left( \frac{n'_g}{n^*_{g,T}} - 1 \right) + \frac{1}{4}\left( \frac{n'_g}{n^*_{g,T}} - 1 \right)^2 \max\left( 1, \left( \frac{n^*_{g,T}}{n'_g} \right)^3 \right) \right)$$

where the first step follows from $R^*_\infty = \frac{\sigma_g^2}{n^*_{g,T}}$, the second step follows from Taylor's inequality applied on the function $x \to \frac{1}{x}$ on $n^*_{g,T}$, and the first step follows from the definition of the infinite norm. By dividing both sides by $R^*_\infty$, rearranging the terms, and taking the max, we obtain

$$\frac{R_\infty(\boldsymbol{n}'_T) - R_\infty(\boldsymbol{n}^*_T)}{R_\infty(\boldsymbol{n}^*_T)} \leq \max_g \left( -\left( \frac{n'_g}{n^*_{g,T}} - 1 \right) + \frac{1}{4}\left( \frac{n'_g}{n^*_{g,T}} - 1 \right)^2 \max\left( 1, \left( \frac{n^*_{g,T}}{n'_g} \right)^3 \right) \right)$$

$$\leq -\min_g \left( \frac{n'_g}{n^*_{g,T}} - 1 \right) + \frac{1}{4} \max_g \left( \frac{n'_g}{n^*_{g,T}} - 1 \right)^2 \max_g \left( \frac{n^*_{g,T}}{n'_g} \right)^3.$$

which completes the proof of Lemma 1.

## C.2 Proof of Theorem 3

**Theorem 3.** *Let $\Sigma_\infty := \sum_{g \in [G]} \sigma_g^2$. For any $\mathcal{D}$ that satisfies Assumptions 1 and 2,*

$$\text{Regret}_{\infty,T}(\text{Variance-UCB}, \mathcal{D}) \leq \left( C_T \sqrt{\Sigma_\infty} + C_T^2 \right) G^{1.5} T^{-1.5} + o(T^{-1.5}) = \tilde{O}(T^{-1.5}).$$

*Proof.* Similarly to Appendix A.5, the total-expectation bounding still holds:

$$\text{Regret}_{\infty,T}(\text{Variance-UCB}, \mathcal{D}) \leq \mathbb{E}[R_\infty(\boldsymbol{n}) - R_\infty^* | \mathcal{A}_T] + \|\boldsymbol{\sigma}^2\|_\infty \mathbb{P}_\pi(\mathcal{A}_T^c). \tag{30}$$

We now upper bound each of the two terms in (30). First, similarly to the case where $p < +\infty$ presented in Appendix A, $\mathbb{P}_\pi(\mathcal{A}_T^c) \leq 2GT^{-2.5}$, so that

$$\|\boldsymbol{\sigma}^2\|_\infty \mathbb{P}_\pi(\mathcal{A}_T^c) \leq 2G\|\boldsymbol{\sigma}^2\|_\infty T^{-2.5} = o(T^{-1.5}). \tag{31}$$

Next, we apply the final lower bound derived in Section B.2 from [13]:

**Lemma 17.** *Conditionally on $\mathcal{A}_T$,*

$$\frac{n_{g,T}}{n_{g,T}^*} - 1 \geq \frac{-G\sqrt{G}}{\sqrt{T}} \left( \frac{C_T}{\sqrt{\Sigma_\infty}} \left(1 + \frac{C_T}{\sqrt{\Sigma_\infty}}\right) + 8\sqrt{2}G^{1/4} \left(\frac{C_T}{\sqrt{\Sigma_\infty}}\right)^{3/4} \sqrt{1 + \frac{C_T}{\sqrt{\Sigma_\infty}}} T^{-3/4} \right).$$

By taking the min over $g \in [G]$ in Lemma 17, the following inequality follows conditionally on $\mathcal{A}_T$:

$$-\min_g \left(\frac{n_{g,T}}{n_{g,T}^*} - 1\right) \leq \frac{G\sqrt{G}}{\sqrt{T}} \left( \frac{C_T}{\sqrt{\Sigma_\infty}} \left(1 + \frac{C_T}{\sqrt{\Sigma_\infty}}\right) + 8\sqrt{2}G^{1/4} \left(\frac{C_T}{\sqrt{\Sigma_\infty}}\right)^{3/4} \sqrt{1 + \frac{C_T}{\sqrt{\Sigma_\infty}}} T^{-3/4} \right). \tag{32}$$

Moreover, since $C_T = \tilde{O}(1)$, the right hand side of the inequality above is in $\tilde{O}(\sqrt{T})$. Next, from Inequality (23), and from $\min_g n_{g,T}^* = T \frac{\min_g \sigma_g^2}{\Sigma_\infty}$,

$$
\begin{aligned}
\max_g \left(\frac{n_{g,T}}{n_{g,T}^*} - 1\right)^2 &\leq \frac{1}{(\min_h n_{h,T}^*)^2} \left( 3G + \frac{4GC_T}{\Sigma_\infty} \left(\min_g \sigma_g + \frac{2C_T}{\sqrt{\min_h n_{h,T}^*}}\right) \sqrt{\min_h n_{h,T}^*} \right)^2 \\
&\leq \frac{18G^2}{(\min_h n_{h,T}^*)^2} + \frac{32G^2 C_T^2}{\Sigma_\infty \min_h n_{h,T}^*} \left(2(\min_g \sigma_g)^2 + 4C_T^2\right) \\
&= \frac{64G^2 C_T^2}{T} \left(1 + \frac{2C_T^2}{\min_g \sigma_g^2}\right) + \frac{18G^2 \Sigma_\infty^2}{T^2 \min_g \sigma_g^4} \\
&= \tilde{O}(T^{-1}),
\end{aligned}
$$

and

$$
\begin{aligned}
\max_g \frac{n_{g,T}^*}{n_{g,T}} &= \frac{1}{\min_g \frac{n_{g,T}}{n_{g,T}^*}} = \frac{1}{1 + \min_g \left(\frac{n_{g,T}}{n_{g,T}^*} - 1\right)} \\
&\leq \frac{1}{\left(1 - \frac{G\sqrt{G}}{\sqrt{T}} \left( \frac{C_T}{\sqrt{\Sigma_\infty}} \left(1 + \frac{C_T}{\sqrt{\Sigma_\infty}}\right) + 8\sqrt{2}G^{1/4} \left(\frac{C_T}{\sqrt{\Sigma_\infty}}\right)^{3/4} \sqrt{1 + \frac{C_T}{\sqrt{\Sigma_\infty}}} T^{-3/4}\right)\right)^+} \\
&= O(1).
\end{aligned}
$$

Combining the previous two inequalities yields

$$\frac{1}{4} \max_g \left(\frac{n_{g,T}}{n_{g,T}^*} - 1\right)^2 \max_g \left(\frac{n_{g,T}^*}{n_{g,}}\right)^3 \leq \frac{\frac{64G^2 C_T^2}{T}\left(1 + \frac{2C_T^2}{\min_g \sigma_g^2}\right) + \frac{18G^2 \Sigma_\infty^2}{T^2 \min_g \sigma_g^4}}{\left[\left(1 - \frac{G\sqrt{G}}{\sqrt{T}}\left(\frac{C_T}{\sqrt{\Sigma_\infty}}\left(1 + \frac{C_T}{\sqrt{\Sigma_\infty}}\right) + 8\sqrt{2}G^{1/4}\left(\frac{C_T}{\sqrt{\Sigma_\infty}}\right)^{3/4}\sqrt{1 + \frac{C_T}{\sqrt{\Sigma_\infty}}}T^{-3/4}\right)\right)^+\right]^3}. \tag{33}$$

Recall that $R_\infty^* = \frac{\Sigma_\infty}{T}$. Applying Lemma 5 on $n_{g,T}$ and using the upper bounds (32) and (33) implies that conditionally on $\mathcal{A}_T$

$$R_\infty(\boldsymbol{n}_T) - R_\infty(\boldsymbol{n}_T^*) \leq \frac{G^{3/2}\Sigma_\infty}{T^{3/2}} \frac{C_T}{\sqrt{\Sigma_\infty}} \left(1 + \frac{C_T}{\sqrt{\Sigma_\infty}}\right)$$

$$+ \frac{G^{7/4}\Sigma_\infty}{T^{9/4}} \left(\frac{C_T}{\sqrt{\Sigma_\infty}}\right)^{3/4} \sqrt{1 + \frac{C_T}{\sqrt{\Sigma_\infty}}}$$

$$+ \frac{\frac{64G^2\Sigma_\infty C_T^2}{T^2}\left(1 + \frac{2C_T^2}{\min_g \sigma_g^2}\right)}{\left[\left(1 - \frac{G\sqrt{G}}{\sqrt{T}}\left(\frac{C_T}{\sqrt{\Sigma_\infty}}\left(1 + \frac{C_T}{\sqrt{\Sigma_\infty}}\right) + 8\sqrt{2}G^{1/4}\left(\frac{C_T}{\sqrt{\Sigma_\infty}}\right)^{3/4}\sqrt{1 + \frac{C_T}{\sqrt{\Sigma_\infty}}}T^{-3/4}\right)\right)^+\right]^3}$$

$$+ \frac{\frac{18G^2\Sigma_\infty^3}{T^3 \min_g \sigma_g^4}}{\left[\left(1 - \frac{G\sqrt{G}}{\sqrt{T}}\left(\frac{C_T}{\sqrt{\Sigma_\infty}}\left(1 + \frac{C_T}{\sqrt{\Sigma_\infty}}\right) + 8\sqrt{2}G^{1/4}\left(\frac{C_T}{\sqrt{\Sigma_\infty}}\right)^{3/4}\sqrt{1 + \frac{C_T}{\sqrt{\Sigma_\infty}}}T^{-3/4}\right)\right)^+\right]^3}.$$

Since the right hand side is deterministic, the inequality above extends by taking the conditional expected value on $\mathcal{A}_T$. Moreover, the second, third, and fourth term are all in $o(T^{-1.5})$, so that:

$$\mathbb{E}[R_\infty(\boldsymbol{n}) - R_\infty^*|\mathcal{A}_T] \leq C_T\sqrt{\Sigma_\infty}\left(1 + \frac{C_T}{\sqrt{\Sigma_\infty}}\right)G^{1.5}T^{-1.5} + o(T^{-1.5}). \tag{34}$$

Finally, combining both inequalities (31) and (34) in Inequality (30) yields

$$\text{Regret}_{\infty,T}(\text{Variance-UCB}, \mathcal{D}) \leq C_T\sqrt{\Sigma_\infty}\left(1 + \frac{C_T}{\sqrt{\Sigma_\infty}}\right)G^{1.5}T^{-1.5} + o(T^{-1.5}),$$

which completes the proof of Theorem 3. $\qquad\square$

### C.3 Proof of Theorem 4

**Lemma 18.** *Let $\boldsymbol{\sigma}^a, \boldsymbol{\sigma}^b$ be two vectors with $\Sigma_\infty^a := \sum_g (\sigma_g^a)^2$, $\Sigma_\infty^b := \sum_g (\sigma_g^b)^2$ and $\Sigma_\infty^a \geq \Sigma_\infty^b$. There exists a $\Sigma_\infty^{a,b} \in [\Sigma_\infty^b, \Sigma_\infty^a]$ such that*

$$d(\boldsymbol{\sigma}^a, \boldsymbol{\sigma}^b) = \frac{\Sigma_\infty^{a,b}}{2T}\|\boldsymbol{\lambda}^*(\boldsymbol{\sigma^a}) - \boldsymbol{\lambda}^*(\boldsymbol{\sigma^b})\|_1$$

*Proof.* Let $\boldsymbol{\sigma}^a, \boldsymbol{\sigma}^b$ be two vectors with $\Sigma_\infty^a := \sum_g (\sigma_g^a)^2$ and $\Sigma_\infty^b := \sum_g (\sigma_g^b)^2$, $\epsilon > 0$, and $\boldsymbol{\lambda} \in [0,1]^G$ such that $\sum_g \lambda_g = 1$. We have:

$$R_\infty(T\boldsymbol{\lambda}; \boldsymbol{\sigma}^a) - R_\infty^*(\boldsymbol{\sigma}^a) = \frac{1}{T}\left(\max_g \frac{\sigma_g^a}{\lambda_g} - \Sigma_\infty^a\right) = \frac{\Sigma_\infty^a}{T}\left(\max_g \frac{\lambda_g^*(\boldsymbol{\sigma^a})}{\lambda_g} - 1\right),$$

therefore,

$$T\boldsymbol{\lambda} \in S_\epsilon^a \iff R_\infty(T\boldsymbol{\lambda}; \boldsymbol{\sigma}^a) - R_\infty^*(\boldsymbol{\sigma}^a) \leq \epsilon$$

$$\iff \frac{\Sigma_\infty^a}{T}\left(\max_g \frac{\lambda_g^*(\boldsymbol{\sigma^a})}{\lambda_g} - 1\right) \leq \epsilon$$

$$\iff \forall g \in [G], \quad \frac{\lambda_g^*(\boldsymbol{\sigma^a})}{\frac{T\epsilon}{\Sigma_\infty^a} + 1} \leq \lambda_g.$$

Hence, the allocation $T\boldsymbol{\lambda}$ is simultaneously in $S_\epsilon^a$ and $S_\epsilon^b$ if and only if

$$\forall g \in [G], \quad \max\left(\frac{\lambda_g^*(\boldsymbol{\sigma^a})}{\frac{T\epsilon}{\Sigma_\infty^a} + 1}, \frac{\lambda_g^*(\boldsymbol{\sigma^b})}{\frac{T\epsilon}{\Sigma_\infty^b} + 1}\right) \leq \lambda_g,$$

In particular, $S_\epsilon^a \cap S_\epsilon^b$ is the polytope

$$\left\{ T\boldsymbol{\lambda} \mid \boldsymbol{\lambda} \in [0,1]^G, \sum_g \lambda_g = 1, \quad \forall g \in [G], \quad \max\left( \frac{\lambda_g^*(\boldsymbol{\sigma^a})}{\frac{T\epsilon}{\Sigma_\infty^a} + 1}, \frac{\lambda_g^*(\boldsymbol{\sigma^b})}{\frac{T\epsilon}{\Sigma_\infty^b} + 1} \right) \leq \lambda_g \right\}$$

which is non-empty if and only if

$$\sum_g \max\left( \frac{\lambda_g^*(\boldsymbol{\sigma^a})}{\frac{T\epsilon}{\Sigma_\infty^a} + 1}, \frac{\lambda_g^*(\boldsymbol{\sigma^b})}{\frac{T\epsilon}{\Sigma_\infty^b} + 1} \right) \leq 1.$$

so that $d(\boldsymbol{\sigma^a}, \boldsymbol{\sigma^b})$ is the (unique) solution to the equation $\sum_g \max\left( \frac{\lambda_g^*(\boldsymbol{\sigma^a})}{\frac{T\epsilon}{\Sigma_\infty^a}+1}, \frac{\lambda_g^*(\boldsymbol{\sigma^b})}{\frac{T\epsilon}{\Sigma_\infty^b}+1} \right) = 1$. To better understand the form of this solution, we introduce the following three decreasing functions:

$$f : \epsilon > 0 \to \sum_g \max\left( \frac{\lambda_g^*(\boldsymbol{\sigma^a})}{\frac{T\epsilon}{\Sigma_\infty^a} + 1}, \frac{\lambda_g^*(\boldsymbol{\sigma^b})}{\frac{T\epsilon}{\Sigma_\infty^b} + 1} \right)$$

$$f_a : \epsilon > 0 \to \sum_g \max\left( \frac{\lambda_g^*(\boldsymbol{\sigma^a})}{\frac{T\epsilon}{\Sigma_\infty^a} + 1}, \frac{\lambda_g^*(\boldsymbol{\sigma^b})}{\frac{T\epsilon}{\Sigma_\infty^a} + 1} \right)$$

$$f_a : \epsilon > 0 \to \sum_g \max\left( \frac{\lambda_g^*(\boldsymbol{\sigma^a})}{\frac{T\epsilon}{\Sigma_\infty^b} + 1}, \frac{\lambda_g^*(\boldsymbol{\sigma^b})}{\frac{T\epsilon}{\Sigma_\infty^b} + 1} \right)$$

First, notice that $d(\boldsymbol{\sigma^a}, \boldsymbol{\sigma^b}) = f^{-1}(1)$. Next, by assuming (WLOG) that $\Sigma_\infty^a \geq \Sigma_\infty^b$, we have $f^a \geq f \geq f^b$, so that $d(\boldsymbol{\sigma^a}, \boldsymbol{\sigma^b}) = f^{-1}(1) \in [(f^b)^{-1}(1), (f^a)^{-1}(1)]$. Moreover,

$$
\begin{aligned}
f_a(\epsilon) &= \sum_g \max\left( \frac{\lambda_g^*(\boldsymbol{\sigma^a})}{\frac{T\epsilon}{\Sigma_\infty^a} + 1}, \frac{\lambda_g^*(\boldsymbol{\sigma^b})}{\frac{T\epsilon}{\Sigma_\infty^a} + 1} \right) \\
&= \frac{\sum_g \max\left( \lambda_g^*(\boldsymbol{\sigma^a}), \lambda_g^*(\boldsymbol{\sigma^b}) \right)}{\frac{T\epsilon}{\Sigma_\infty^a} + 1} \\
&= \frac{\frac{1}{2}\sum_g \lambda_g^*(\boldsymbol{\sigma^a}) + \lambda_g^*(\boldsymbol{\sigma^b}) + |\lambda_g^*(\boldsymbol{\sigma^a}) - \lambda_g^*(\boldsymbol{\sigma^b})|}{\frac{T\epsilon}{\Sigma_\infty^a} + 1} \\
&= \frac{1 + \frac{1}{2}\|\boldsymbol{\lambda}^*(\boldsymbol{\sigma^a}) - \boldsymbol{\lambda}^*(\boldsymbol{\sigma^b})\|_1}{\frac{T\epsilon}{\Sigma_\infty^a} + 1},
\end{aligned}
$$

where the first step stems from the definition of $f_a$, the second step stems from $\max(x,y) = \frac{x+y+|x-y|}{2}$, the third step stems from $\sum_g \lambda_g^* = 1$, and the fourth step stems from the defintion of the $1-$norm. The last equation implies

$$f_a^{-1}(1) = \frac{\Sigma_\infty^a}{2T} \|\boldsymbol{\lambda}^*(\boldsymbol{\sigma^a}) - \boldsymbol{\lambda}^*(\boldsymbol{\sigma^b})\|_1.$$

Similarly, $f_b^{-1}(1) = \frac{\Sigma_\infty^b}{2T} \|\boldsymbol{\lambda}^*(\boldsymbol{\sigma^a}) - \boldsymbol{\lambda}^*(\boldsymbol{\sigma^b})\|_1$. Therefore:

$$\frac{\Sigma_\infty^b}{2T} \|\boldsymbol{\lambda}^*(\boldsymbol{\sigma^a}) - \boldsymbol{\lambda}^*(\boldsymbol{\sigma^b})\|_1 \leq d(\boldsymbol{\sigma^a}, \boldsymbol{\sigma^b}) \leq \frac{\Sigma_\infty^a}{2T} \|\boldsymbol{\lambda}^*(\boldsymbol{\sigma^a}) - \boldsymbol{\lambda}^*(\boldsymbol{\sigma^b})\|_1,$$

which concludes the proof. $\qquad \square$

**Theorem 4.** *For any online policy $\boldsymbol{\pi}$, there exists an instance $\mathcal{D}_{\boldsymbol{\pi}}$ such that for any $T \geq 1$,*

$$Regret_{\infty,T}(\boldsymbol{\pi}, \mathcal{D}_{\boldsymbol{\pi}}) \geq \frac{1}{2} G^{1.5} T^{-1.5}.$$

*Proof.* The proof consists of constructing two instances that optimize the trade-off stated in Lemma 6. The first instance, which is denoted $\mathcal{D}^a$, consists of $G$ groups of standard normal distributions. Next, select a group $h$ such that $\mathbb{E}_{\boldsymbol{\pi}, \mathcal{D}^a}[n_{g,T}]$ is minimal. Notice that by minimality of $\mathbb{E}_{\boldsymbol{\pi}, \mathcal{D}^a}[n_{h,T}]$,

$$\mathbb{E}_{\pi,\mathcal{D}^a}[n_{h,T}] \leq \frac{1}{G} \sum_{g \in [G]} \mathbb{E}_{\pi,\mathcal{D}^a}[n_{g,T}] = \frac{1}{G}\mathbb{E}_{\pi,\mathcal{D}^a}\left[\sum_{g \in [G]} n_{g,T}\right] = \frac{T}{G}. \tag{35}$$

The second instance, which is denotes $\mathcal{D}^b$, is defined as follows:

$$\mathcal{D}_g^b = \begin{cases} \mathcal{N}(0, 1+\nu) & \text{For } g = h \\ \mathcal{N}(0, 1) & \text{Otherwise} \end{cases}$$

We have

$$\begin{cases} \boldsymbol{\sigma}^a &= (1, \ldots, 1) \\ \boldsymbol{\sigma}^b &= \boldsymbol{\sigma}^a + \nu e_h \end{cases}$$

so that $\Sigma_\infty^a = G$, $\Sigma_\infty^b = G + \nu$ and

$$\boldsymbol{\lambda}^*(\boldsymbol{\sigma}^a) = \frac{1}{G}(1, \ldots, 1)$$

$$\forall g \in [G], \quad \lambda_g^*(\boldsymbol{\sigma}^b) = \begin{cases} \frac{1}{G+\nu} & \text{For } g \neq h \\ \frac{1+\nu}{G+\nu} & \text{For } g = h \end{cases}$$

We introduce the following Lemma, which proof is deferred to later in this section.

**Lemma 19.** *The following inequality holds:* $\sum_{g \in [G]} \mathbb{E}_{\boldsymbol{\pi},\mathcal{D}^a}[n_{g,T}]KL(\mathcal{D}_g^a||\mathcal{D}_g^b) \leq \frac{T\nu^2}{2G}$.

By combining both Lemmas 18 and 19 in the inequality stated in Lemma 6, we obtain

$$\max\left\{\text{Regret}_\infty(\boldsymbol{\pi}, \mathcal{D}^a), \text{Regret}_\infty(\boldsymbol{\pi}, \mathcal{D}^b)\right\} \geq \frac{\min(\Sigma_\infty^a, \Sigma_\infty^b)}{2T}\|\boldsymbol{\lambda}^*(\boldsymbol{\sigma}^a) - \boldsymbol{\lambda}^*(\boldsymbol{\sigma}^b)\|_1 \exp\left(-\frac{T\nu^2}{2G}\right)$$

$$= e^{-\frac{T\nu^2}{2G}}\frac{G}{2T}\left(\frac{1}{G} - \frac{1}{G+\nu} + (G-1)\left(\frac{1+\nu}{G+\nu} - \frac{1}{G}\right)\right)$$

$$= \frac{e^{-\frac{T\nu^2}{2G}}}{2}\frac{G}{T} \times \frac{(G+\nu) - G + (G-1)(G+G\nu - G - \nu)}{G(G+\nu)}$$

$$= \frac{e^{-\frac{T\nu^2}{2G}}}{2}\frac{\nu G}{T} \times \frac{1 + (G-1)^2}{G(G+\nu)}.$$

By setting $\nu = \sqrt{\frac{G}{T}} \leq 1$ and $\mathcal{D}_{\boldsymbol{\pi}} := \text{argmax}\left\{\text{Regret}_\infty(\boldsymbol{\pi}, \mathcal{D}^a), \text{Regret}_\infty(\boldsymbol{\pi}, \mathcal{D}^b)\right\}$, the previous inequality implies

$$\text{Regret}_\infty(\boldsymbol{\pi}, \mathcal{D}_{\boldsymbol{\pi}}) = \max\left\{\text{Regret}_\infty(\boldsymbol{\pi}, \mathcal{D}^a), \text{Regret}_\infty(\boldsymbol{\pi}, \mathcal{D}^b)\right\} \geq \frac{e^{-1/2}}{2}\frac{G^{1.5}}{T^{1.5}}\frac{1 + (G-1)^2}{G(G+1)} \geq \frac{1}{2}G^{1.5}T^{-1.5}$$

which completes the proof of Theorem 4, . $\qquad\square$

We now prove Lemma 19.

**Lemma 19.** *The following inequality holds:* $\sum_{g \in [G]} \mathbb{E}_{\boldsymbol{\pi},\mathcal{D}^a}[n_{g,T}]KL(\mathcal{D}_g^a||\mathcal{D}_g^b) \leq \frac{T\nu^2}{2G}$.

*Proof.* For convenience, we set $\nu := \sqrt{\frac{G}{T}} < 1$. The formula for the $KL-$divergence of two univariate normal distributions of zero mean implies

$$KL(\mathcal{D}_h^a||\mathcal{D}_h^b) = \frac{1}{2}\left(\log\left(\frac{\sigma^2(1+\nu)}{\sigma^2}\right) + \frac{\sigma^2 - (\sigma^2(1+\nu))}{\sigma^2(1+\nu)}\right).$$

The taylor expansion of the expression above can be derived by combining the expansions of both the functions $x \to \log(1 + x)$ and $x \to \frac{1}{1+x}$ in the domain $(0, 1)$:

$$\frac{1}{2}\left(\log\left(\frac{\sigma^2(1+\nu)}{\sigma^2}\right) + \frac{\sigma^2 - (\sigma^2(1+\nu))}{\sigma^2(1+\nu)}\right) = \frac{1}{2}\left(-\sum_{k\geq 1}\frac{(-1)^k}{k}\nu^k - \nu\sum_{k\geq 0}(-1)^k\nu^k\right)$$

$$= \frac{1}{2}\sum_{k\geq 1}(-1)^k\nu^k\left(1 - \frac{1}{k}\right)$$

$$= \frac{\nu^2}{2}\sum_{k\geq 0}(-1)^k\nu^k\left(1 - \frac{1}{k+2}\right)$$

$$\leq \frac{\nu^2}{2},$$

where the first step stems from the inequality $n_T \leq T$, the second step stems from using the previous two equalities, the third step stems from simplifying the infinite sum term-wise, the fourth step stems from the expansion of a geometric serie, and the fifth step stems from $\mu = \frac{1}{\sqrt{T}} \geq 0$. Since $\mathcal{D}^a$ and $\mathcal{D}^b$ have the same distribution at all coordinates except coordinate $h$, we have:

$$\sum_{g\in[G]}\mathbb{E}_{\boldsymbol{\pi},\boldsymbol{\mathcal{D}}^a}[n_{g,T}]KL(\mathcal{D}_g^a||\mathcal{D}_g^b) = \mathbb{E}_{\boldsymbol{\pi},\boldsymbol{\mathcal{D}}^a}[n_{h,T}]KL(\mathcal{D}_h^a||\mathcal{D}_h^b) \leq \frac{T\nu^2}{2G}$$

where the inequality follows from the minimality of $\mathbb{E}_{\boldsymbol{\pi},\boldsymbol{\mathcal{D}}^a}[n_{h,T}]$ and $\sum_g \mathbb{E}_{\boldsymbol{\pi},\boldsymbol{\mathcal{D}}^a}[n_{g,T}] = T$. This concludes the proof. $\qquad\square$

