# OpenReview forum: "An active learning framework for multi-group mean estimation"
_NeurIPS.cc/2023/Conference — NeurIPS 2023 poster_

### Official Review · Reviewer_kPNK · 2023-07-04

**Soundness:** 2 fair
**Presentation:** 3 good
**Contribution:** 2 fair
**Rating:** 5
**Confidence:** 4

**Summary:**

This manuscript studies a special type of bandit problem: instead of maximizing the rewards, the learner aims to minimize the L_p norm of the variance vector for the mean estimators of each arm. The motivation of this problem is multi-group mean estimation, where a small total variance is desired.

The authors proposed a variance-UCB algorithm, which maintains an upper confidence bound for the variance of each arm, and then chooses the arm which maximizes a certain quantity determined by its current UCB and number of pulls. This quantity is motivated by the expression of optimal allocation in hindsight. This algorithm achieves a regret of O(T^{-2}) for all finite p, which is optimal by matching lower bounds. For p = infinity, this algorithm recovers the \Theta(T^{-1.5}) optimal regret obtained by [Antos et al. 2008, Carpentier et al. 2011].

**Strengths:**

The problem setting is practically relevant, and the authors proposed a novel form of UCB-type algorithm to solve it. The resulting regret bound is also tight.

**Weaknesses:**

I have several major concerns about this work:

1. The target L_p norm in Eqn. (1) is not properly motivated. The authors should at least describe a scenario where minimizing this specific quantity is meaningful. For example, is p = 1 or infty the only interesting scenario? Can the current result be extended to some value of p < 1, say p = 1/2 (I believe this is relevant when one uses absolute estimation error).

2. Although the algorithm is a new UCB-type algorithm, both the intuition and analysis are very straightforward. By the KKT condition for the objective, the optimal allocation should give the same value to all arms for a certain quantity; so the algorithm simply pulls the arm with the largest quantity in order to reduce it. The analysis is then standard, possibly with the exception of Lemma 8 (the key part of the new UCB algorithm). However, this contribution alone does not reach the bar for a NeurIPS publication.

3. Very importantly, the current manuscript does not have a tight regret dependence on the number of arms G. Obtaining so would require a more careful analysis in both the upper and lower bounds. In particular, the lack of tight dependence on G makes the current two-point lower bound much less interesting.

4. There seems to be an issue in the upper bound analysis. Note that Lemma 3 only proves a one-sided inequality, i.e. an upper bound on n - n^*. However, when the authors applied Lemma 4, a two-sided upper bound on |n - n^*| is needed, and this requires additional explanation. I understand this seems to be fine because the sum of n_g is always T, but arguing in this way would have an additional factor of G which I don't know if that's necessary, and in my opinion (see above) the right dependence on G is important.

Also, if one would only like to obtain an upper bound on n - n^*, the complicated arguments below Lemma 8 seem unnecessary. It seems that applying Lemma 8 to the last time a certain group g is chosen should be enough.

Additional comments:

Page 3: "the optimization program (3) can be NP-hard to prove". Please add a justification.

Page 4, expression of C_T: the precise form is not interesting at all. Just say C_T = C * log(T) for a large enough constant C.

Page 18, Eqn. (21): where does Sigma_p come from? Lemma 2 does not involve this term.

**Questions:**

I'll likely increase my rating if the authors could work out the tight regret dependence on G.

---

> ### Author Rebuttal · Authors · 2023-08-09
>
> Thank you for your detailed comments on our paper.
>
> Q1 (Lp norms):
> The expression of the $p$-norm for $p \geq 1$ is $\|\boldsymbol{x}\|_p := (x_1^p + \ldots + x_n^p)^{1/p}$. The most frequent measures of error we find in the ML literature are $p = 1$ (absolute error), $p=2$ (squared error), and $p=\infty$ (worst-case error). General $p$-norms are less frequent in practice, but their study in this case allows us to connect results across $p = 1, 2, +\infty$, thus providing a framework for the most three common measurement errors. We believe $p=2$ is also interesting as it prevents ``outliers'', no variances of any group's mean estimator will be too large, and it provides a nice trade-off between the sum of all the variances ($p=1$) and the worst-case variance ($p=\infty$).
>
> For $p < 1$,  $\|.\|_p$ is no longer a norm, as it violates the triangle inequality. Maybe we misunderstood the comment: is it possible to provide more detail for the $p = 1/2$ case and how it relates to absolute estimation error?
>
> Q2 (algorithm and analysis): We believe that the algorithm is indeed intuitive, which is a desirable feature, although the analysis is far from immediate.
> Lemma 8 is only a first step to solving the first problem, as it gives us a good control of the last pulled arm, but information about the previous arms is lost. Lemma 9 uniformly bounds the number of pulls with a quantity that is decoupled from the algorithm, while Lemmas 10 and 11 are necessary for the tightness of the bounding.
>
> We also require using a Taylor expansion argument to deal with the complex curvature of our objective function, which does not arise in the traditional MAB setting. This has the benefit of giving the tightest first order approximation. (This is also why we don't have large constants in front of the regret, as opposed to most bandits regret bounds).
>
> We remark that our results address open questions in [Carpentier et al. 2011], which directly asked if one can relax the Gaussian assumption to remove the dependency on $\sigma_{min}$, and also asked if tight lower bounds can be obtained.
>
> Q3 (dependence on $G$): You are indeed correct that the dependence on $G$ in the regret bounds is interesting, as well as the smallest variance $\sigma_{min}$ and total sum of variances $\Sigma_{\infty}$. Upon more careful analysis for the case where $p=\infty$, we are able to show an upper bound on the order of $\Sigma_{\infty} G^{1.5} T^{-1.5}$ and a matching lower bound of $\Sigma_{\infty} G^{1.5} T^{-1.5}$. This is a significant improvement in the $G$ and $\sigma_{\min}$ parameters, compared to the upper bound result of [Carpentier et al. 2011] for the case of sub-Gaussian distributions, where they provided a bound on the order of $\Sigma_\infty \sigma_{\min}^{-1} G^{2.5}   T^{-1.5}$. We will add details of these more refined results to the paper.
>
> Q4 (upper bound analysis):
> The proof we submitted gives sub-optimal dependencies on $G$, and the reason is the one that you pointed out (the naive bounding $n - n^* \leq ||n-n^*||$ is sub-optimal). We did so because we were more focused on the dependency of $T$ in our bound. However, we can now derive a tight dependency on $G$, as we mentioned above. Here is a brief explanation on how to do so: first, we combine the upper bound in Lemma 3 with a lower bound derived from similar arguments found in [Carpentier et al. 2011]. Next, for $p$ finite, we approximate the objective with a quadratic function in $(n - n^*)$, and maximize it subject to the two-sided bound on $n - n^*$. For $p=\infty$, we can rewrite the objective function to get a more tractable expression in $n - n^*$.
>
> Responses to additional comments:
>
> -To simplify presentation, we will remove the mention of NP-hardness from the paper.
> -We provided the exact expression of $C_T$ for implementation
>  and replicability purposes.
>
> -$\Sigma_p$ is introduced in line 172. The upper bound in Lemma 2 involves $n$, while the upper bound in Lemma 3 involves $n^*$.
>
> Carpentier, A., Lazaric, A., Ghavamzadeh, M., Munos, R., & Auer, P. (2011, October). Upper-confidence-bound algorithms for active learning in multi-armed bandits. In International Conference on Algorithmic Learning Theory (pp. 189-203). Berlin, Heidelberg: Springer Berlin Heidelberg.

---

> > ### Comment · Reviewer_kPNK · 2023-08-14
> >
> > Thank you for the detailed comments. It's great to see that the optimal dependence on $G$ could be obtained for the case $p=\infty$, and I'll happily increase my score. Just two additional questions:
> >
> > 1. For $p=\infty$, does your current lower bound technique already give you the right dependence on $G$, or you need a different argument?
> >
> > 2. For $p\in (1,\infty)$, what is currently your best upper and lower bound in terms of the dependence on $G$?

---

> > > ### Author Response · Authors · 2023-08-18
> > >
> > > 1-The new lower bound proof uses the same technique, but with a better adversarial instances to get the right dependency on $G$. Instead of using $2$ instances, we use $G+1$ instances.
> > >
> > > 2-For the case where $p$ is finite, we have not yet derived a (good) lower and upper bound that depends on all the parameters yet (including $G$). We conjecture that the same roadmap for $p=\infty$ should yield a tight upper bound (mainly using double bounds on $n - n^*$ with more careful approximations on the objective). We also conjecture that, for the lower bound, the same $G+1$ instances with more careful analysis of the dissimilarity function $d(\cdot,\cdot)$ will yield a good result. We hope to update our paper in the future with this extra analysis, but cannot guarantee concrete results for the finite $p$ scenario that depend on $G$. For $p=\infty$, we are happy to report tight lower and upper bounds that depend on $G$, as discussed in the earlier post

---

### Official Review · Reviewer_JGPv · 2023-07-06

**Soundness:** 4 excellent
**Presentation:** 3 good
**Contribution:** 3 good
**Rating:** 7
**Confidence:** 3

**Summary:**

This paper focuses on the active learning algorithm for multi-group mean estimation. The authors focus on minimizing the $l_p$-norm of the variance vector. This paper proposes the variance-UCB algorithm to actively select which group to sample in each round. The sample complexities for $p<\infty$ and $p=\infty$ are provided, and the tightness of proposed algorithm on $T$ is also verified.

**Strengths:**

This paper focuses on the novel problem of estimating the mean values of multiple groups with minimal variance. The authors adopt $l_p$-norm to measure the variances of group mean estimates. The variance-UCB algorithm is designed and analyzed for different values of $p$. In addition, the lower bound is established to justify the tightness of the proposed algorithm on the dependency of $T$. The simulation results are provided to corroborate the theoretical findings.

**Weaknesses:**

1. The authors mainly focus on the dependency of $T$. However, the variance of each group also influences the sample complexity. It will be helpful to derive the upper bound that reflects these instance-dependent quantities.

2. The limitation part is missing in the paper.

**Questions:**

The questions are provided in the weaknesses section.

**Limitations:**

The authors adequately addressed the limitations

---

> ### Author Rebuttal · Authors · 2023-08-09
>
> We thank you for your time reviewing our paper, and for your supportive comments. Below we address the two questions raised in your review.
>
> Q1 (Instance-dependent upper bound): Our primary focus in the analysis was dependence on $T$, which is the parameter that we envision growing large in most practical applications. However, our analysis also allows us to provide tight (upper and lower) bounds for all parameters, including $G$ and $\sigma$. For example, for $p = +\infty$, we can provide a refined upper bound of $\Sigma_{\infty}G^{1.5} T^{-1.5} + o(T^{-1.5})$, and a refined lower bound in $\Sigma_{\infty}G^{1.5} T^{-1.5}$. We will add these results to the final version.
>
> Q2 (limitations): We will add a brief limitations section to the final version of our paper. We will include discussions of the limitations arising from our two modeling assumptions, and possible future directions that may arise from removing these assumptions.

---

> > ### Comment · Reviewer_JGPv · 2023-08-10
> >
> > Thank you very much for addressing these concerns.

---

### Official Review · Reviewer_kmyz · 2023-07-06

**Soundness:** 2 fair
**Presentation:** 3 good
**Contribution:** 3 good
**Rating:** 5
**Confidence:** 3

**Summary:**

This paper propose the Variance-UCB algorithm to sequentially learn the mean in a multigroup setting in order to minimize the variance over all mean estimates, and prove the regret of the algorithm is optimal for both finite and infinite p values.

**Strengths:**

1. The Variance-UCB algorithm in this paper automatically achieves optimal regret for both cases when $p$ is finite and $p=\infty$, and provide solid theoretical proofs for both the general lower bound and the matching regret for the algorithm.

2. The authors support their statements with empirical results by varying different parameters in time horizon $T$, norm parameter $p$, group numbers $G$ and sub-Gaussian parameters.

3. The paper is in general well-written.

**Weaknesses:**

The experiment results lack the comparison with other benchmark results in the literature. The authors only mention that the varying lowest variance has no effect on the regret results when p is infinite which is a known result in the literature.

One typo: line 65 "it is thekkir variances..." should it be "their"?

**Questions:**

As stated in the weakness, can the authors compare the Variance-UCB algorithm to other results in the literature? For example, the authors mention in the paper that this work lies in the frame work of a multi-arm bandits setting, but other bandit works in this line did not take care of the variances of the mean estimates. It would be good to empirically show that the proposed algorithm outperforms these bandit algorithms in the literature in this task.

**Limitations:**

Lack of comparison to other benchmark algorithms in the literature.

---

> ### Author Rebuttal · Authors · 2023-08-09
>
> Thank you for your review and for your kind words about our paper. Thank you also for pointing out the typo; we will correct that.
>
>
> 1- Comparison with other algorithms:
>
> For the case where the norm $p = +\infty$, two algorithms are known: [Antos et al. 2008, Carpentier et al. 2011]. [Carpentier et al. 2011] provides an algorithm (B-AS) and has already shown that it outperforms the algorithm (GAFS-MAX) derived in [Antos et al. 2008]. Our algorithm (Variance-UCB) coincides with (B-AS) when $p = +\infty$. We chose not to repeat the experiment made in [Carpentier 2011].
>
> For the case where $p < +\infty$, we provide the first algorithm for this setting to the best of our knowledge.
>
> Applying without any adapting traditional algorithms (Thompson sa,pling or $\epsilon$-greedy) would yield to sub-optimal solutions. In our setting, the optimal strategy samples all groups asymptotically. It is not clear how to adopt traditional bandit algorithms that, at the minimum, converge asymptotically to the right solution.
>
> 2- Experiments around the smallest variance:
>
>  [Antos 2008, Carpentier 2011] derive an upper bound that depends on $\sigma_{\min}^{-1}$ when $p=\infty$. While [Carpentier 2011] showed that the smallest variance has no effect when the distributions are exactly Gaussian, they did not show this for the general sub-Gaussian case which our paper does.
> Our experiments simply verify our theoretical result that the smallest variance does not impact the regret significantly when $p=\infty$ and the distributions are sub-Gaussian.
>
> In stark contrast, we show that when $p$ is finite the smallest variance does still have an effect, even for Gaussian distributions.
>
> András Antos, Varun Grover, and Csaba Szepesvári. Active learning in multi-armed bandits.307
> pages 287–302, 10 2008.
>
> Carpentier, A., Lazaric, A., Ghavamzadeh, M., Munos, R., & Auer, P. (2011, October). Upper-confidence-bound algorithms for active learning in multi-armed bandits. In International Conference on Algorithmic Learning Theory (pp. 189-203). Berlin, Heidelberg: Springer Berlin Heidelberg.

---

> > ### Comment · Reviewer_kmyz · 2023-08-14
> >
> > Thank you for your response, and I will keep my score.

---

### Official Review · Reviewer_m8Me · 2023-07-07

**Soundness:** 3 good
**Presentation:** 3 good
**Contribution:** 3 good
**Rating:** 7
**Confidence:** 4

**Summary:**


This paper studies the mean estimation problem under the multi-armed bandit setting. Here, we have a group of populations (random variables) with unknown mean and standard deviations. The goal is to estimate the mean of each group on the fly and optimize the regret (measured by different kinds of norms). Their major contribution is a group of confidence interval based online learning algorithms and showing the optimality of these algorithms.


**Strengths:**

I think many working on multi-armed bandits and statistical learning have naturally wondered this question. I am glad to see this result and that existing UCB type algorithms still work in the new setting.


**Weaknesses:**

The text in intro feels a bit strange to me (e.g., quite a bit of empty sells and and “analyst” also sounds odd to me). I think most people understand the importance of this problem.

It would be helpful if the authors could relate this result to Stein estimator type research and stratified sampling. They feel related.


**Questions:**

I have a few questions:

1. The baseline considers a relaxed version of the original problem because the latter is NP-Hard. Nevertheless, the regrets are tight. Does that imply a certain kind of integrality gap bound between the NP-hard original problem and the relaxed problem?
2. How is the regret related to the number of populations, and is this also tight?
3. Is it possible to comment on your result with the long line of research related to Stein estimators? It seems that those statistics results also aim to estimate population mean via shrinkage methods.
4. Is there a reason to call a random variable a population, e.g., they have finite populations, or they are inconsequential naming?
5. Are there speculations and educated guesses on the generalization of the problem, e.g., what if my goal is to estimate the standard deviation/second moment of each population?

---

> ### Author Rebuttal · Authors · 2023-08-09
>
> Thank you for your review of our work and for your kind words about our contributions to the bandits and statistical learning literature. Below we address the questions posed in your review.
>
> Q1 (integrality gap): Indeed, our tightness results are with respect to the original problem, which implies that the integrality gap goes to zero as $T$ grows large.
>
> Q2 (dependence on $G$): Our main focus in the analysis was dependence on $T$, since this is the parameter that we envision growing large in practice. Upon more careful analysis for the case where $p=\infty$, we are able to show an upper bound on the order of $\Sigma_{\infty} G^{1.5} T^{-1.5}$ and a matching lower bound of $\Sigma_{\infty} G^{1.5} T^{-1.5}$. This is a significant improvement in the $G$ and $\sigma_{\min}$ parameters, compared to the upper bound result of [Carpentier et al. 2011] for the case of sub-Gaussian distributions, where they provided a bound on the order of $\Sigma_\infty \sigma_{\min}^{-1} G^{2.5}   T^{-1.5}$. We will add details of these more refined results to the paper.
>
> Q3 (Stein estimators): Stein estimators are biased, and we exclusively want to focus on unbiased which is why we look at the sample mean for each group. A key motivation at looking at the norm of the variances of the mean estimators is to make sure we are sampling each group enough, motivated by fairness reasons. Thus, using a biased estimator (such as the Stein method) is problematic in our setting since we care about fairness across groups.
>
> Q4 (population terminology): We use the term ``population'' in relation to a survey methodology setting, where the decision-maker/analyst/surveyor decides where to collect data in a dynamic fashion. However, we assume each group can be sampled as many times as we want, so essentially the populations is infinite in this setting and the naming is inconsequential.
>
> Q5 (extensions to other statistics): We conjecture that our results and algorithmic framework would generalize to  other well-defined statistics, as long as one can construct an estimator with high probability concentration guarantees.
>
> Carpentier, A., Lazaric, A., Ghavamzadeh, M., Munos, R., & Auer, P. (2011, October). Upper-confidence-bound algorithms for active learning in multi-armed bandits. In International Conference on Algorithmic Learning Theory (pp. 189-203). Berlin, Heidelberg: Springer Berlin Heidelberg.

---

### Official Review · Reviewer_ZDPZ · 2023-07-07

**Soundness:** 3 good
**Presentation:** 4 excellent
**Contribution:** 3 good
**Rating:** 7
**Confidence:** 3

**Summary:**

The paper provides a general bandit-based active learning approach to the problem of learning the means of several disjoint groups so as to optimize for the variance of the resulting estimates as measured by the p-norm of the variance vector. An algorithm is proposed which samples based on a certain upper confidence bound designed for the variance p-norm, and its regret guarantees are studied. The authors identify and carefully study two regimes: p < infty and p = infty, and prove a somewhat surprising, and tight, dichotomy: namely, the regret is only Theta(1/T^2) for p < infty but increases to Theta(1/T^1.5) for p = infty. The tight lower bounds is achieved via a simple and explicit lower-bounding construction, whereas the upper bounds follow from two different appropriate upper bounds on the curvature of the variance vector p-norm. Some experiments are given to confirm the theoretically identified tight rates.

**Strengths:**

The main strength of the paper is that it presents a very clean and natural UCB-based algorithm which, for reasons that are transparently shown via a sequence of intuitive arguments (such as a simple Taylor-based curvature bound on the p-norm vector), achieves the tight rates for the important problem of identifying group means with small variance using as few samples as possible. I also consider it quite important and strong that the authors were able to locate the corresponding tight lower bounds, which is achieved by a surprisingly clean and simple construction (where two incompatible worlds are provided which are identical except for two groups' distributions that are slightly tweaked). In this manner, this simple case of optimizing for p-norms of the variance vector becomes, in a sense, resolved (up to potential tweaks to the setup which --- as mentioned in the future work section --- could involve constraining the sampling policies etc.)

Another strength is the lucid and careful presentation of the results --- the paper is generally quite well-written (aside from a few typos that I encourage the authors to locate and fix).

**Weaknesses:**

In the theory part, there are in my opinion no big weaknesses --- the case is worked out quite comprehensively. Of course, a (small) weakness of the approach is the elusive nature of the subgaussian parameters that the proof assumes --- some experiments are later run to convince the reader that you don't necessarily need to know these constants exactly, but the intricate ways in which they are present in the bounds make the situation not as pleasant to deal with theoretically (on which note, I would like to ask the authors to --- for full transparency ---- more prominently mention the assumption, e.g. in the contributions section).

Another relatively small weakness is that the experimental section could be tightened up, in the sense of some plots being expanded to be more informative and some off-the-cuff remarks elaborated on. (See the Questions section below.) Still, no major problems there in my opinion.

**Questions:**

First, for the theory part, and for the experimental part alike, I would like to better understand the nature of requiring the smallest variance to be nonzero as well as it featuring in the performance guarantees and --- in fact, as mentioned on line 275 --- somewhat paradoxically leading to higher regret as it decreases. If there is an intuitive explanation or some examples for why this should be so, I'd like to learn more.

Secondly, could the authors plot out some other norms between l2 and linfty? This way the transition effect as p -> infty could be better portrayed and understood.

Thirdly, can plots such as Figure 2 be extended to make the convergence of some "difficult" curves (such as C=0.001 in that example) more visible?

Finally, do the authors have any insights into why the transition in Figure 4, even though it takes a long time to wait for it, kicks in so rapidly? Thanks!

---

> ### Author Rebuttal · Authors · 2023-08-09
>
> Thank you for your review and for your kind words about our paper and results.
>
> We will add a brief discussion about the sub-Gaussianity assumption in the contributions.
>
> For the "relatively small weakness" in the experimental section, corresponding to Q2 and Q3, we will incorporate your feedback to plot some other norms between $\ell_2$ and $\ell_\infty$ and  extend the x-axis on Figure 2 to better show convergence of the $C=0.001$ case.
>
> Q1 (Requiring non-zero variance): The smallest variance appears in the regret bounds (although not in the dominant term for the case of $p=\infty$). Suppose the unknown distributions are Bernoulli. Intuitively, an analyst cannot  distinguish (using finitely many samples) between a group with arbitrarily small variance and a group with zero variance, but the optimal strategy can be quite different in these two cases.
>
> Q4 (Rapid convergence in Figure 4):
> To better understand why this happens, we ran a similar experiment for $p = 1, 2, +\infty$, which we will add to the paper. Initially the algorithm samples (on average) uniformly across all groups due to the UCB constant outweighing the sample variance estimates, and each group must wait to be sampled enough times for the algorithm to estimate its optimal sampling rate. This delay will naturally increase with the number of groups. Once we are in the right range, the algorithm samples the highest variances first. This causes abrupt variations in the case where $p$ is small because the objective function is very sensitive to changes in one coordinate.

---

> > ### Comment · Reviewer_ZDPZ · 2023-08-16
> > **Acknowledgment**
> >
> > Thank you for your response to my questions. In particular, with the help of the authors' answers, I now better understand the empirical properties of the proposed method (e.g. through the explanation about the rapid convergence region), and expect and encourage the authors to add such intuitions, as well as revised plots, to the updated version of the manuscript. I will happily keep my score and positive opinion of this work.
> >
> > I also enjoyed reading the discussions with the other reviewers --- in particular, appreciating their point that working out optimal dependence on the number of groups would be quite appealing in this setting. I look forward to seeing the added proof of the new tight bound in the revised manuscript.

---

### Decision · Program_Chairs · 2023-09-21

**Decision:**

Accept (poster)

**Comment:**

Reviewers were generally positive about the problem studied and the technical contribution of this paper. There were some concerns about the regret dependence on $G$.  The authors' rebuttal answered this question satisfactory. In particular, it is good to see that the optimal dependence on $G$ could be obtained for the case $p = \infty$